# Derandomized Online-to-Non-convex Conversion for Stochastic Weakly Convex Optimization

**Fanfan Ji & Xiao-Tong Yuan**[*]
School of Intelligence Science and Technology
Nanjing University
Suzhou, 215163, China
State Key Laboratory for Novel Software Technology
Nanjing University
Nanjing, 210023, China
`{jiff1995,xtyuan}@nju.edu.cn`

## Abstract

Online-to-non-convex conversion (O2NC) is an online updates learning framework for producing Goldstein $(\delta, \epsilon)$-stationary points of non-smooth non-convex functions with optimal oracle complexity $\mathcal{O}(\delta^{-1}\epsilon^{-3})$. Subject to auxiliary *random interpolation or scaling*, O2NC recapitulates the stochastic gradient descent with momentum (SGDM) algorithm popularly used for training neural networks. Randomization, however, introduces deviations from practical SGDM. So a natural question arises: Can we derandomize O2NC to achieve the same optimal guarantees while resembling SGDM? On the negative side, the general answer is *no* due to the impossibility results of Jordan et al. (2023), showing that no dimension-free rate can be achieved by deterministic algorithms. On the positive side, as the primary contribution of the present work, we show that O2NC can be naturally derandomized for *weakly convex* functions. Remarkably, our deterministic algorithm converges at an optimal rate as long as the weak convexity parameter is no larger than $\mathcal{O}(\delta^{-1}\epsilon^{-1})$. In other words, the stronger stationarity is expected, the higher non-convexity can be tolerated by our optimizer. Additionally, we develop a periodically restarted variant of our method to enable more progressive updates when the iterates are far from stationarity. The resulting algorithm, which can be viewed as a momentum-restarted variant of SGDM, has been empirically demonstrated to be effective and efficient for training ResNet and ViT models.

## 1 Introduction

Classic machine learning optimization methods often rely crucially on convexity and/or smoothness assumptions to guarantee the convergence to optima (Nesterov, 2018; Bubeck, 2015; Snyman, 2005). However, many modern large-scale machine learning models, such as residual neural networks (He et al., 2016) and transformers (Vaswani et al., 2017), involve non-smooth and non-convex objective functions. These models achieve state-of-the-art performance precisely thanks to their capability to learn highly complex, nonlinear hidden representations in data. Given such widespread use, efficient non-smooth and non-convex optimization algorithms are of central theoretical and practical interest.

Specifically, this paper is concerned with stochastic gradient algorithms for solving the following expected risk minimization problem ubiquitous in statistical learning:

$$\min_{w \in \mathbb{R}^d} R(w) := \mathbb{E}_{Z \sim \mathcal{D}}[\ell(w; Z)], \tag{1}$$

where $\ell : \mathbb{R}^d \times \mathcal{Z} \mapsto \mathbb{R}^+$ is a non-negative loss function whose value $\ell(w; z)$ measures the loss evaluated at $z \in \mathcal{Z}$ with parameter $w \in \mathbb{R}^d$ and $\mathcal{D}$ represents a distribution over the measurable

---

[*]Corresponding author.

set $\mathcal{Z}$. We consider the setting where the loss $\ell$ is Lipschitz continuous with respect to its first argument, yet potentially neither convex nor smooth. In contrast to the smooth counterpart, finding an $\epsilon$-stationary point (or even a neighborhood around it) of a non-smooth objective in 1 is generally intractable (Zhang et al., 2020; Kornowski & Shamir, 2022). This intractability motivates the employment of Goldstein $(\delta, \epsilon)$-stationarity (see Definition 1) as a notion of approximate convergence for non-smooth non-convex functions (Zhang et al., 2020). The study of efficient algorithms with finite-time complexity guarantees for finding $(\delta, \epsilon)$-stationary points has since received ever emerging interests in non-smooth, non-convex optimization (Zhang et al., 2020; Davis et al., 2022; Cutkosky et al., 2023; Jordan et al., 2023; Tian & So, 2024; Kong & Lewis, 2025).

Developed by Cutkosky et al. (2023), the online-to-non-convex conversion (O2NC) method finds $(\delta, \epsilon)$-stationary points of problem 1 with $\mathcal{O}(\delta^{-1}\epsilon^{-3})$ stochastic gradient oracle calls, attaining the optimal first-order complexity. As outlined in Algorithm 1, O2NC essentially converts an online learner (highlighted in light red) to a stochastic gradient optimizer (highlighted in light blue). It recursively updates the increments $\Delta_n := w_n - w_{n-1}$ between consecutive iterates via invoking an online convex optimization (OCO) algorithm $\mathcal{A}$ to minimize the regret $\sum_{n=1}^{N} \langle \hat{g}_n, \Delta_n - \Delta \rangle$, where the stochastic subgradient $\hat{g}_n$ is evaluated at a random intermediate state $v_n = w_{n-1} + s_n \Delta_n$ with $s_n \sim \text{Uniform}([0, 1])$. The optimal oracle complexity can be achieved by any instantiation of $\mathcal{A}$ with an optimal regret bound, such as online gradient descent (OGD) (Zinkevich, 2003).

Beyond theoretical optimality, another appealing feature of the O2NC framework is its potential to recover the stochastic momentum-based optimizers widely used in neural network training. Indeed, under the random interpolation scheme over iterates, O2NC equipped with projected OGD becomes a clipped variant of SGD with momentum (SGDM) Cutkosky et al. (2023). Alternatively, Zhang & Cutkosky (2024) introduced the Exponentiated O2NC (E-O2NC) framework with exponential random scaling on the updates, which nearly recovers the standard SGDM when combined with unconstrained online mirror descent (OMD) (Beck & Teboulle, 2003). Within the O2NC framework, Ahn et al. (2024) showed that the widely used Adam optimizer (Kingma & Ba, 2015) corresponds to a discounted variant of the Follow-the-Regularized Leader (FTRL) algorithm (Gordon, 1999), and analyzed the merits of its key algorithmic components: momentum and discounting factors.

---

**Algorithm 1:** Online-to-non-convex Conversion (O2NC) (Cutkosky et al., 2023)

**Input** : OCO algorithm $\mathcal{A}$, $K, T \in \mathbb{N}$, initial point $w_0$ and increment $\Delta_1$. Set $N = K \times T$.

**for** $n = 1, ..., N$ **do**

> /* Stochastic gradient optimizer */
> Update $w_n = w_{n-1} + \Delta_n$;
> Compute random interpolation $v_n = w_{n-1} + s_n \Delta_n$, where $s_n \sim \text{Uniform}([0, 1])$;
> Randomly sample $z_n \sim \mathcal{D}$ and obtain $\hat{g}_n \in \partial \ell(v_n; z_n)$;
>
> /* Online learning of increment */
> Send the linear loss $\langle \hat{g}_n, \Delta \rangle$ to $\mathcal{A}$ and receive the next increment $\Delta_{n+1}$ from $\mathcal{A}$.

**end**

Set $w_t^{(k)} = w_{(k-1)T+t}, \forall k \in [K], t \in [T]$, and $\bar{w}^{(k)} = \frac{1}{T} \sum_{t=1}^{T} w_t^{(k)}$.

**Output:** $\bar{w}_T \sim \text{Uniform}(\{\bar{w}^{(k)} : k \in [K]\})$.

---

Despite the appeal of O2NC in explaining the effectiveness and efficiency of SGDM-style optimizers, the resulting algorithmic correspondence inherently relies on several auxiliary randomization procedures, such as uniform interpolation on iterates or exponential scaling on increments. However, such randomization components are rarely, if ever, used in practical implementations of SGDM. This fundamental gap motivates us to investigate the following question:

> *Can the O2NC technique be derandomized to still achieve optimal dimension-free guarantees while maintaining a close resemblance to SGDM in the non-smooth, non-convex setting?*

Unfortunately, the general answer to the above question is *negative*: it has been shown by Jordan et al. (2023, Theorem 1) that no dimension-free rate can be achieved by deterministic algorithms in the worst case. Fortunately, in this paper, we show that for weakly convex functions, one can indeed develop deterministic variants of O2NC that find Goldstein-stationary solutions with optimal rates.

## 1.1 OVERVIEW OF OUR RESULTS

Our main contribution is a derandomized O2NC framework (Algorithm 2) for solving the stochastic optimization problem 1 with a $\rho$-weakly convex risk function, meaning that $R(\cdot) + \frac{\rho}{2}\|\cdot\|^2$ is convex. Inspired by the original O2NC, our key development leverages weak convexity to naturally reformulate the optimization of iterates $w_n$ as the online learning of increments $\Delta_n$ with respect to quadratic losses $\langle \hat{g}_n, \cdot \rangle + \frac{\gamma}{2}\|\cdot\|^2$ for some $\gamma \geq \rho$. In contrast to evaluating gradients at a random intermediate point $v_n$ between consecutive iterates $w_n$ and $w_{n-1}$, our method evaluates gradients exactly at each iterate $w_n = w_{n-1} + \Delta_n$, yielding a fully deterministic algorithm (up to stochastic gradient estimation). Concretely, we propose two optional online learners for updating the increments $\Delta_n$, which are summarized as follows:

**Derandomized O2NC with clipped OGD (Section 3.2).** The first option is a simple projected OGD algorithm under an appropriate ball constraint. The resulting algorithm can be interpreted as a clipped variant of SGDM, without requiring extra random interpolations. Our convergence analysis (Corollary 1) shows that the proposed deterministic method identifies a $(\delta, \epsilon)$-stationary point with $\mathcal{O}\left(\delta^{-1}\epsilon^{-3} + \rho^3\delta^2 + \delta^{-1}\right)$ stochastic gradient oracle calls, which is dominated by the optimal rate $\delta^{-1}\epsilon^{-3}$. Strikingly, the weak convexity parameter $\rho$ does not appear in this dominant term, and it can scale as large as $\mathcal{O}(\delta^{-1}\epsilon^{-1})$ in its associated term before matching the optimal rate. This observation implies that the smaller the required $(\delta, \epsilon)$, the higher the non-convexity our optimizer can tolerate while still achieving optimal oracle complexity.

**Derandomized O2NC with periodically restarted OGD (Section 3.3).** As in the original O2NC, our first approach based on OGD under an explicit ball constraint tends to be overly conservative for increment updates and is also impractical from the viewpoint of standard SGDM implementation. To overcome this limitation, as our second scheme, we further introduce a novel periodically restarted OGD procedure, which *resets the increments to zero after a fixed number of iterations*. The resulting method is nearly identical to the standard SGDM, differing only in that the momentum update is periodically restarted from scratch. Under a novel notion of $(\mu, \epsilon)$-regularized stationarity (see Definition 2), which is equivalent to the Goldstein stationarity, we establish in Corollary 2 that the proposed deterministic, unconstrained O2NC algorithm converges with the composite rate $\mathcal{O}\left(\mu^{1/2}\epsilon^{-7/2} + \rho^{7/3}\mu^{-2/3} + \mu^{1/2}\right)$, where $\rho = \mathcal{O}(\mu^{1/2}\epsilon^{-3/2})$ is admissible without dominating the optimal term $\mu^{1/2}\epsilon^{-7/2}$. Coupled with our theoretical results, we conduct extensive numerical experiments on standard benchmark tasks to validate that the proposed momentum-restarted SGDM variant achieves performance comparable to, or even better than, the standard SGDM when training deep residual networks and vision transformers (Section 4).

## 1.2 RELATED WORK

Our work is situated within the broad landscape of non-smooth and non-convex optimization. Below we provide a brief overview of closely relevant prior work.

**Non-smooth optimization.** The foundations of non-smooth optimization trace back to the seminal work of Clarke (1975); Goldstein (1977). While asymptotic analysis of non-smooth optimization problems has been extensively studied (Benaïm et al., 2005; Davis et al., 2020; Bolte & Pauwels, 2021), non-asymptotic guarantees for generic non-smooth settings remained elusive for a long time. A pivotal advance was made by Zhang et al. (2020), who established finite-time complexity bounds for subgradient algorithms targeting Goldstein stationary points—this breakthrough spurred a surge of follow-up research (Davis et al., 2022; Kornowski & Shamir, 2022; Cutkosky et al., 2023; Jordan et al., 2023; Kornowski & Shamir, 2024). Notably, building on the online-to-batch conversion paradigm (Cesa-Bianchi et al., 2004), Cutkosky et al. (2023) introduced the O2NC framework, which for the first time derived optimal convergence rates for stochastic non-smooth, non-convex optimization. By instantiating different online learners within O2NC, one can recover several widely used optimizers: SGDM corresponds to online mirror descent (Zhang & Cutkosky, 2024), Adam aligns with a variant of Follow-the-Regularized Leader (Ahn & Cutkosky, 2024), and most recently, Ahn et al. (2025) showed that a generalized O2NC framework also encompasses schedule-free SGD (Defazio et al., 2024).

**Stochastic weakly convex optimization.** The class of weakly convex functions, first introduced in English by Nurminskii (1973), is broad and easily identifiable, as it encompasses all composition

forms $h \circ c$ of convex functions and smooth maps. For weakly convex optimization problems, a vast literature of asymptotic convergence results has been established for stochastic optimization algorithms (Ermol'ev & Norkin, 1998; Duchi & Ruan, 2018). However, finite-time non-asymptotic convergence rates remained largely unresolved until a series of breakthroughs by Davis & Grimmer (2019); Davis & Drusvyatskiy (2019); Mai & Johansson (2020), who showed that various SGD/SGDM algorithms achieve the optimal $\mathcal{O}(\epsilon^{-4})$ rate for producing an $\epsilon$-stationary point of the Moreau envelope of the objective function. From a variational analysis perspective, several distinct notions of approximate subdifferentials have been analyzed and compared for weakly convex functions (van Ackooij et al., 2024). In practice, weakly convex optimization has found widespread applications in deep learning, signal processing and control theory (see, e.g., Duchi & Ruan, 2018; Davis & Drusvyatskiy, 2019; Drusvyatskiy & Paquette, 2019; Pougkakiotis & Kalogerias, 2023).

## 2 PRELIMINARIES

Let us begin by formally introducing some notation, key assumptions, and preliminary results on non-smooth and non-convex optimization.

### 2.1 NOTATION AND ASSUMPTIONS

**Notation.** Throughout this paper, we denote $\|\cdot\|$ as the Euclidean norm, and $\langle\cdot,\cdot\rangle$ as the Euclidean inner product. For a vector set $V \subseteq \mathbb{R}^d$, we denote $\text{dist}(0,V) := \inf_{v \in V} \|v\|$ and $\text{conv}\{V\}$ the convex hull of $V$. For any positive integer $N$, we abbreviate $[N] = \{1, ..., N\}$. The symbol $\mathbb{B}_\delta(w)$ denotes the closed ball of radius $\delta$ centered on $w$, and $\text{clip}_D(w) := w \min\left\{1, \frac{D}{\|w\|}\right\}$ denotes the Euclidean projection operator associated with the constraint of $\mathbb{B}_D(0)$. For a pair of functions $f, f' \geq 0$, we use $f = \mathcal{O}(f')$ to denote $f \leq cf'$ for some universal constant $c > 0$.

We say that a function $f : \mathbb{R}^d \mapsto \mathbb{R}$ is $G$-Lipschitz continuous if $|f(w) - f(w')| \leq G\|w - w'\|$ for all $w, w' \in \mathbb{R}^d$. The Clarke subdifferential (Clarke, 1990) of a non-smooth function $f$ at $w \in \mathbb{R}^d$ is denoted by $\partial f(w)$. Recall that $f$ is said to be $\rho$-weakly convex for some $\rho > 0$ if the quadratically regularized function $f(\cdot) + \frac{\rho}{2}\|\cdot\|^2$ is convex, or equivalently

$$f(w) \geq f(w') + \langle g', w - w'\rangle - \frac{\rho}{2}\|w - w'\|^2, \ \ \forall w, w' \in \mathbb{R}^d, g' \in \partial f(w').$$

A key class of weakly convex functions takes the composite form $f(x) = h(c(x))$, where $h$ is a convex and $G$-Lipschitz continuous function, and $c$ is a smooth mapping with a $L$-Lipschitz Jacobian. These composite functions are neither smooth nor convex, but rather $GL$-weakly convex (Davis & Drusvyatskiy, 2019). A concrete example, as considered in our experimental study, is neural networks equipped with smooth activation functions (e.g., softplus and GeLU): their loss functions follow the composite form $h \circ c$, where $h$ denotes a convex top-layer predictor (e.g., cross-entropy loss) and $c$ represents a smooth hierarchical feature mapping. For additional examples of weakly convex functions, we refer interested readers to Davis & Drusvyatskiy (2019); Asi & Duchi (2019).

The Moreau envelope (Rockafellar, 1997) of a $\rho$-weakly convex function $f$ with parameter $\lambda \in (0, \rho^{-1})$ is defined by $f_\lambda(w) := \inf_{u \in \mathbb{R}^d}\left\{f(u) + \frac{1}{2\lambda}\|u - w\|^2\right\}$, and the associated proximal mapping is given by $\text{prox}_{\lambda f}(w) := \arg\min_{u \in \mathbb{R}^d}\left\{f(u) + \frac{1}{2\lambda}\|u - w\|^2\right\}$. The following result (Böhm & Wright, 2021) characterizes the continuous differentiability of Moreau envelope functions.

**Lemma 1.** *Let $f$ be a $\rho$-weakly convex function and $\lambda \in (0, \rho^{-1})$ be a scalar. Then the Moreau envelope $f_\lambda$ is continuously differentiable with gradient $\nabla f_\lambda(w) = \frac{1}{\lambda}\left(w - \text{prox}_{\lambda f}(w)\right) \in \partial f(\text{prox}_{\lambda f}(w))$, which is $L$-Lipschitz continuous with parameter $L = \max\left\{\lambda^{-1}, \frac{\rho}{1-\rho\lambda}\right\}$.*

**Assumptions.** We next impose some key assumptions on the loss and risk functions in problem 1 for stochastic gradient-based optimization.

**Assumption 1.** *For any $z \in \mathcal{Z}$, the loss function $\ell(\cdot; z)$ is $G$-Lipschitz with respect to its first argument. Moreover, the expected risk function $R$ is $\rho$ weakly-convex.*

**Assumption 2** (Stochastic oracle). *For each $w \in \mathbb{R}^d$, it holds that $\ell'(w) = \mathbb{E}_{Z \sim \mathcal{D}}[\ell'(w; Z)] \in \partial R(w)$, where $\ell'(w; z) \in \partial\ell(w; z)$ for any $z \in \mathcal{Z}$.*

Also, we assume that $R^* = \min_{w \in \mathbb{R}^d} R(w) > -\infty$ and abbreviate $\Delta R_0 := R(w_0) - R^*$.

## 2.2 REGULARIZED GOLDSTEIN STATIONARITY CRITERION

For general non-smooth, non-convex functions, Goldstein $(\delta, \epsilon)$-stationarity (Goldstein, 1977) serves as a standard criterion for convergence analysis, and is defined formally below.

**Definition 1** $((\delta, \epsilon)$-Stationarity). *The Goldstein $\delta$-subdifferential of a Lipschitz function $f$ at a point $w \in \mathbb{R}^d$ is the convex hull of all Clarke subgradients at points in a $\delta$-ball around $w$, i.e.,*

$$\partial_\delta f(w) := conv \left\{ \cup_{v \in \mathbb{B}_\delta(w)} \partial f(v) \right\}.$$

*A point $w$ is called a $(\delta, \epsilon)$-stationary point if $dist\,(0, \partial_\delta f(w)) \leq \epsilon$.*

While finite-time guarantees on the $(\delta, \epsilon)$-stationarity have been thoroughly studied in the original O2NC framework (Cutkosky et al., 2023), the corresponding analysis inherently requires online increment updates to be explicitly constrained within a ball of radius $\delta\epsilon^2$ which can be overly conservative. Inspired by Zhang & Cutkosky (2024), we next introduce a novel regularized variant of $(\delta, \epsilon)$-stationarity which obviates the need for such explicit constraints, thereby enabling potentially more aggressive increment updates. Given a subset $V \subseteq \mathbb{R}^d$, we define the set-valued mapping $\partial_V f := \text{conv}\{\cup_{v \in V} \partial f(v)\}$. We further define the $\mu$-regularized subdifferential norm as

$$\|\partial f(w)\|_{+\mu} := \inf_{V \subseteq \mathbb{R}^d} \left\{ \text{dist}(0, \partial_V f) + \mu \sup_{v \in V} \|v - w\|^2 \right\}. \tag{2}$$

**Definition 2** $((\mu, \epsilon)$-Regularized Stationarity). *A point $w$ is said to be a $(\mu, \epsilon)$-regularized stationary point of a Lipschitz function $f$ if $\|\partial f(w)\|_{+\mu} \leq \epsilon$.*

**Remark 1.** *Intuitively, $(\mu, \epsilon)$-regularized stationarity jointly controls the magnitude of the convex hull of subgradients evaluated at points in an underlying subset $V$, as well as the proximity of $V$ to $w$. In contrast to the relaxed Goldstein stationarity introduced by Zhang & Cutkosky (2024), our formulation employs a supremum norm penalty instead of its average counterpart, which yields an exact equivalence to the original $(\delta, \epsilon)$-stationarity, as summarized in the following lemma (see Appendix A.1 for its proof).*

**Lemma 2.** *Let $\delta, \epsilon, \mu > 0$ be arbitrary positive values. Consider a Lipschitz function $f$.*

*(a) If $w$ is a $(\delta, \epsilon)$-stationary point, then it is also a $\left(\frac{\epsilon}{\delta^2}, 2\epsilon\right)$-regularized stationary point.*

*(b) If $w$ is a $(\mu, \epsilon)$-regularized stationary point, then it is also a $\left(\sqrt{\frac{\epsilon}{\mu}}, \epsilon\right)$-stationary point.*

We further state the following lemma, which establishes the monotonicity of $\|\partial f(w)\|_{+\mu}$ with respect to the regularization strength parameter $\mu$. A proof is provided in Appendix A.2.

**Lemma 3.** *Let $f$ be a Lipschitz function. Then for any $w \in \mathbb{R}^d$ and $0 < \mu_1 \leq \mu_2$, it holds that $\|\partial f(w)\|_{+\mu_1} \leq \|\partial f(w)\|_{+\mu_2}$.*

## 3 DERANDOMIZED O2NC FOR WEAKLY CONVEX OPTIMIZATION

Building on the O2NC framework, we develop in this section a derandomized stochastic subgradient method for computing Goldstein-style stationary points of weakly convex functions. An overview of our algorithm is presented in Section 3.1. Notably, the online learning module of our algorithm incorporates two alternative subroutines for updating the increments: clipped OGD and periodically restarted OGD, which are described and analyzed in detail in Sections 3.2 and 3.3, respectively.

## 3.1 ALGORITHM

The pseudocode for our Derandomized Online-to-non-convex Conversion (D-O2NC) algorithm is presented in Algorithm 2. In contrast to the original O2NC (Algorithm 1), the stochastic gradient optimizer module (highlighted in light blue ) of our algorithm removes the random interpolation step $v_n = w_{n-1} + s_n\Delta_n$ and instead directly evaluates the subgradient $\hat{g}_n$ at the current iterate $w_n = w_{n-1} + \Delta_n$. For the online learning module (highlighted in light red ), we propose two alternative OGD variants for updating the next increment $\Delta_{n+1}$, both are designed to minimize regret over a sequence of quadratic losses of the form $\langle \hat{g}_n, \cdot \rangle + \frac{\gamma}{2}\| \cdot \|^2$, as elaborated below:

---

**Algorithm 2:** Derandomized O2NC

---

**Input** : $\gamma, \eta > 0$, $D > 0$ (optional), $K, T \in \mathbb{N}$, initial $w_0$ and $\Delta_1 = 0$. Set $N = K \times T$.

**for** $n = 1, ..., N$ **do**

> /* Stochastic gradient optimizer */
> Update $w_n = w_{n-1} + \Delta_n$;
> Randomly sample $z_n \sim \mathcal{D}$ and compute $\hat{g}_n \in \partial\ell(w_n; z_n)$;

> /* Online learning of increments */
> **(Option-I)** Update $\Delta_{n+1} = \text{clip}_D\left[(1 - \eta\gamma)\Delta_n - \eta\hat{g}_n\right]$;/* Clipped OGD */
> **(Option-II)** /* Periodically restarted OGD */
> **if** $\text{mod}(n + 1, T) \not\equiv 1$ **then**
> > Update $\Delta_{n+1} = (1 - \eta\gamma)\Delta_n - \eta\hat{g}_n$;
> **end**
> **else**
> > Set $\Delta_{n+1} = 0$;
> **end**

**end**

Set $w_t^{(k)} = w_{(k-1)T+t}, \forall k \in [K], t \in [T]$, and $\bar{w}^{(k)} = \frac{1}{T}\sum_{t=1}^{T} w_t^{(k)}$.

**Output:** $\bar{w}_T \sim \text{Uniform}(\{\bar{w}^{(k)} : k \in [K]\})$.

---

- **Option-I (clipped OGD):** The online learner $\mathcal{A}$ is instantiated as a projected OGD iteration $\Delta_{n+1} = \text{clip}_D\left[(1 - \eta\gamma)\Delta_n - \eta\hat{g}_n\right]$ with learning rate $\eta$ over a $D$-ball constraint.
- **Option-II (periodically restarted OGD):** We adopt an unconstrained OGD iteration $\Delta_{n+1} = (1 - \eta\gamma)\Delta_n - \eta\hat{g}_n$, with the modification that $\Delta_{n+1}$ is reset to 0 whenever $\text{mod}(n+1, T) \equiv 1$. In other words, the OGD update for $\Delta_n$ is restarted from scratch after every $T$ steps of iteration.

Inspired by the original O2NC, our algorithm minimizes a sequence of quadratic losses in the online setting with the following motivation: for a $\rho$-weakly convex risk function $R$ and any $\gamma \geq \rho$, we have $R(w_n) - R(w_{n-1}) \leq \mathbb{E}\left[\langle\hat{g}_n, \Delta_n\rangle + \frac{\gamma}{2}\|\Delta_n\|^2\right]$. This implies that the increments $\Delta_n$ can be selected sequentially to minimize the regret term $\sum_{n=1}^{N}\langle\hat{g}_n, \Delta_n\rangle + \frac{\gamma}{2}\|\Delta_n\|^2$, thereby ensuring that the function value gap $R(w_N) - R(w_0)$ admits a tight upper bound. Further details on the regret guarantees of OGD for online quadratic loss minimization are provided in Appendix C.

## 3.2 RESULTS FOR D-O2NC UNDER CLIPPED OGD

Recall that in Option-I, the increments are updated via $\Delta_{n+1} = \text{clip}_D\left[(1 - \eta\gamma)\Delta_n - \eta\hat{g}_n\right]$ which is a projected OGD iteration over the quadratic loss function $\langle\hat{g}_n, \Delta\rangle + \frac{\gamma}{2}\|\Delta\|^2$ subject to a $D$-ball constraint. Notably, this update step can be connected to the SGDM method popularly used for training deep learning models (Sutskever et al., 2013; Cutkosky & Orabona, 2019).

**Recovering SGDM.** By setting $m_n = -\gamma\Delta_n$ and $\beta = \eta\gamma$, we can rephrase the Option-I update as

$$w_n = w_{n-1} - \gamma^{-1}m_n;$$
$$m_{n+1} = \text{clip}_D\left[(1 - \beta)m_n + \beta\hat{g}_n\right].$$

This procedure corresponds to a clipped variant of SGDM, where $m_n$ denotes the search direction (constrained to lie within a $D$-ball), $\hat{g}_n$ is the stochastic subgradient, $\gamma$ is the learning rate, and $\beta$ is the momentum parameter. Compared to the clipped SGDM formula implied by the original O2NC (Cutkosky et al., 2023), our variant eliminates random perturbations on the iterates entirely.

**Complexity guarantees.** The following theorem establishes the convergence rate of Algorithm 2 for finding $(\delta, \epsilon)$-stationary points of weakly convex functions. See Appendix B.2 for its proof.

**Theorem 1.** *Suppose that Assumption 1 and Assumption 2 hold. Let $\gamma \geq \rho$ be an arbitrary scalar, $\eta \leq \frac{1}{8\gamma}$, $K, T$ be positive integers, and $D$ be an arbitrary positive number. Then for any $\delta \geq TD$, the sequence $\{\bar{w}^{(k)}\}_{k=1}^{K}$ generated by Algorithm 2 with Option-I satisfies*

$$\mathbb{E}\left[\frac{1}{K}\sum_{k=1}^{K} dist(0, \partial_\delta R(\bar{w}^{(k)}))\right] \leq \frac{\eta G^2}{D} + \left(\gamma T + \frac{2}{\eta}\right)\frac{D}{T} + \frac{G}{\sqrt{T}} + \frac{\Delta R_0}{DKT}.$$

As a direct consequence of Theorem 1, the following corollary establishes the complexity of Algorithm 2 (Option-I) for finding Goldstein $(\delta, \epsilon)$-stationary points. A proof of this corollary is provided in Appendix B.3.

**Corollary 1.** *Suppose that Assumption 1 and Assumption 2 hold. Let $\delta, \epsilon > 0$ denote the target Goldstein stationarity parameters, and $N$ be the total iteration budget. We set*

$$T = \left\lceil (\delta N)^{2/3} \right\rceil, \quad K = \left\lfloor \frac{N}{T} \right\rfloor, \quad \gamma = \frac{N^{1/3}}{\delta^{2/3}}, \quad \eta = \frac{1}{8N}, \quad D = \frac{\delta^{1/3}}{N^{2/3}}.$$

*Suppose further that*

$$N \geq \frac{(G^2 + G + \Delta R_0 + 17)^3}{\delta \epsilon^3} + \rho^3 \delta^2 + \frac{1}{\delta}.$$

*Then the output $\bar{w}_T$ by Algorithm 2 with Option-I satisfies*

$$\mathbb{E}\left[ dist\left(0, \partial_\delta R(\bar{w}_T)\right)\right] \leq \epsilon.$$

**Remark 2.** *We comment that the $\mathcal{O}(\delta^{-1}\epsilon^{-3})$ rate, which dominates the composite complexity bound of Corollary 1, is indeed tight for weakly convex functions. The key insight is that the $\mathcal{O}(\delta^{-1}\epsilon^{-3})$ rate is known to be optimal for all $\epsilon \leq \mathcal{O}(\delta)$ (Cutkosky et al., 2023)—a result that holds even for smooth functions, let alone their superclass of weakly convex functions.*

**Remark 3.** *Notably, the weak convexity parameter $\rho$ does not appear in the dominant rate term, but instead in a suboptimal term $\rho^3 \delta^2$ which allows it to scale up to $\mathcal{O}(\delta^{-1}\epsilon^{-1})$ without dominating the optimal rate. In other words, as higher convergence precision is required, our algorithm can tolerate a larger weak convexity parameter while preserving the optimal complexity rate.*

**Remark 4.** *The hyperparameter choices specified in Corollary 1 are uniquely determined by the iteration budget $N$ and and the target stationarity precisions $\delta, \epsilon$, which can typically be user-specified in practical applications. For example, setting $\delta = \epsilon = \mathcal{O}(N^{-1/4})$, Corollary 1 yields $T = \mathcal{O}(N^{1/2})$, $K = \mathcal{O}(N^{1/2})$, $\gamma = \mathcal{O}(N^{1/2})$, $\eta = \mathcal{O}(N^{-1})$, and $D = \mathcal{O}(N^{-3/4})$.*

### 3.3 RESULTS FOR D-O2NC UNDER PERIODICALLY RESTARTED OGD

While Algorithm 2 with Option-I achieves optimal dimension-free iteration complexity, the clipped OGD iteration restricts the increments $\Delta_n$ to lie within a sufficiently small ball—a constraint that may be overly conservative. To address this limitation, we further propose a novel periodically restarted OGD procedure (Option-II) for the OCO module of our algorithm. Specifically, for each time step $n \geq 1$, we update $\Delta_{n+1} = (1 - \eta\gamma)\Delta_n - \eta\hat{g}_n$, and reset $\Delta_{n+1} = 0$ whenever $\mathrm{mod}(n + 1, T) \equiv 1$. This unconstrained OGD procedure allows for more progressive updates, particularly when iterates are far from stationarity.

**Remark 5.** *Regarding the design of the OCO module, Algorithm 2 with Option-II aligns with E-O2NC (Zhang & Cutkosky, 2024) in spirit, as the latter instantiates its OCO module with an unconstrained OMD procedure. Despite this similarity, our algorithm differs from E-O2NC in two key respects: (1) ours is deterministic and eliminates the need for random scaling of increments; (2) our algorithm avoids both exponential weighting of subgradients in loss construction and exponential moving average (EMA) of iterates for output generation—making it more amenable to practical implementation.*

**Remark 6.** *It is worth highlighting that the proposed periodically restarted OGD procedure can be readily extended to the original O2NC framework for general non-smooth, non-convex optimization under the well-behavedness assumption (Cutkosky et al., 2023). This holds because, under this assumption, quadratic loss functions of the form $\langle \hat{g}_n, \cdot \rangle + \frac{\gamma}{2} \|\cdot\|^2$ (with arbitrary $\gamma > 0$) can similarly be constructed for O2NC (or E-O2NC).*

**Recovering SGDM.** A notable consequence of adopting the periodically restarted OGD procedure is that we can explicitly rewrite the update rule for Algorithm 2 (Option-II) as

$$w_n = w_{n-1} - \gamma^{-1} m_n;$$
$$m_{n+1} = ((1 - \beta)m_n + \beta\hat{g}_n) \, \mathbf{1}_{\{\mathrm{mod}(n+1,T) \neq 1\}},$$

where $m_n := -\gamma\Delta_n$, $\beta := \eta\gamma$, and $\mathbf{1}_{\{\cdot\}}$ denotes the indicator function. The above update rule is nearly identical to standard SGDM, with the only distinction being that the update of the search

direction $m_n$ is reset to zero after every $T$ rounds of iteration. A similar connection to SGDM was also established for the E-O2NC method (Zhang & Cutkosky, 2024), albeit under more sophisticated algorithmic designs (see Remark 5 for further discussion).

**Complexity guarantees.** The following is our main result on the convergence rate of Algorithm 2 with periodically restarted OGD (Option-II). A proof of this result is provided in Appendix B.4.

**Theorem 2.** *Suppose that Assumption 1 and Assumption 2 hold. Let $\gamma \geq \rho$ be an arbitrary scalar, $\eta \leq \frac{1}{8\gamma}$, $K, T$ be positive integers, and $D$ be an arbitrary positive number. Then for any $\mu \leq \frac{\gamma}{8DT^2}$, the sequence $\{\bar{w}^{(k)}\}_{k=1}^{K}$ generated by Algorithm 2 with Option-II satisfies*

$$\mathbb{E}\left[\frac{1}{K}\sum_{k=1}^{K}\left\|\partial R(\bar{w}^{(k)})\right\|_{+\mu}\right] \leq \frac{\eta G^2}{D} + \left(\gamma T + \frac{1}{\eta}\right)\frac{D}{T} + \frac{G}{\sqrt{T}} + \frac{\Delta R_0}{DKT}.$$

**Remark 7.** *Unlike in Theorem 1, where $D$ serves as a hyperparameter in Option-I, the scalar $D$ in Theorem 2 does not appear in Option-II: it is introduced solely for analytical purposes.*

Building on Theorem 2, we further establish the following result regarding the complexity of Algorithm 2 (Option II) for finding $(\mu, \epsilon)$-regularized stationary points. See Appendix B.5 for its proof.

**Corollary 2.** *Suppose that Assumption 1 and Assumption 2 hold. Let $\mu, \epsilon > 0$ denote the target regularized stationarity parameters and $N$ be the total iteration budget. We set*

$$T = \left\lceil N^{4/7}\mu^{-2/7}\right\rceil, \quad K = \left\lfloor \frac{N}{T}\right\rfloor, \quad \gamma = N^{3/7}\mu^{2/7}, \quad \eta = \frac{1}{8N}.$$

*Suppose further that*

$$N \geq \frac{(4G^2 + 32\Delta R_0 + 1)^{7/2}\mu^{1/2}}{\epsilon^{7/2}} + \frac{\rho^{7/3}}{\mu^{2/3}} + \mu^{1/2}.$$

*Then the output $\bar{w}_T$ of Algorithm 2 with Option-II satisfies*

$$\mathbb{E}\left[\|\partial R(\bar{w}_T)\|_{+\mu}\right] \leq \epsilon.$$

**Remark 8.** *By virtue of Lemma 2, setting $\mu = \delta^{-2}\epsilon$ yields that the bound in Corollary 2 implies a complexity of $\mathcal{O}\left(\delta^{-1}\epsilon^{-3} + \rho^{7/3}\delta^{4/3}\epsilon^{-2/3} + \delta^{-1}\epsilon^{1/2}\right)$ for producing $(\delta, \epsilon)$-stationary points, which is dominated by the optimal term $\delta^{-1}\epsilon^{-3}$. Consistent with the discussion in Remark 3, the weak convexity parameter $\rho$ can scale up to $\mathcal{O}(\mu^{1/2}\epsilon^{-3/2})$ without dominating the optimal component.*

## 3.4 COMPARISON WITH PRIOR RESULTS

In Table 1, we summarize the complexity bounds and key properties of D-O2NC, and compare them to those of several other subgradient-based methods for weakly convex optimization, including SGD (Davis & Drusvyatskiy, 2019), SGDM (Mai & Johansson, 2020) and Interpolated Normalized Gradient Descent (INGD) (Davis et al., 2022). We make the following key observations:

- **Comparison with INGD.** Our D-O2NC is deterministic (up to the use of stochastic oracles) and achieves dimension-free, optimal complexity with respect to $(\delta, \epsilon)$-stationarity. In contrast, INGD is randomized by design and difficult to extend to the stochastic setting; additionally, its corresponding complexity bound is dimension-dependent, though it exhibits a sharper dependence on $\rho$ and $(\delta, \epsilon)$.

- **Comparison with SGD and SGDM.** The reported optimal complexity of $\mathcal{O}(\epsilon^{-4})$ for SGD and SGDM corresponds to the $\epsilon$-stationarity of Moreau envelope, i.e., $\|\nabla R_{1/\bar{\rho}}\| \leq \epsilon$ with $\bar{\rho} = \mathcal{O}(\rho)$. While our optimal bounds are not directly comparable to this complexity due to the distinct criteria employed, we highlight the following key observations: 1) Lemma 1 establishes that an $\epsilon$-stationary point of the Moreau envelope implies a $(\epsilon/(2\rho), \epsilon)$-stationary point of the original objective (see Remark 11 in Appendix D for details); accordingly, the bounds for SGD and SGDM imply an $\mathcal{O}(\delta^{-1}\epsilon^{-3})$ complexity for finding $(\delta, \epsilon)$-stationary points, albeit under the relatively restrictive choice $\delta = \epsilon/(2\rho)$; 2) As shown in Theorem 3 (Appendix D), a $(\delta, \epsilon)$-stationary point implies an $(\epsilon + \sqrt{\delta})$-stationary point of the Moreau envelope, which however

yields a suboptimal complexity of $\mathcal{O}(\epsilon^{-5})$ for achieving $\epsilon$-stationarity (Corollary 3); 3) For second-order smooth functions, building on the result of Cutkosky et al. (2023, Proposition 15) it can be readily shown that D-O2NC recovers the optimal $\mathcal{O}(\epsilon^{-3.5})$ complexity for finding $\epsilon$-stationary point, which however cannot be automatically implied by the tabulated results for SGD/SGDM. Finally, the weak convexity parameter $\rho$ can scale up to $\mathcal{O}(\delta^{-1}\epsilon^{-1})$ in our bounds without dominating the optimal rate, a feature not shared by those of SGD and SGDM.

| Method | $(\delta, \epsilon)$-stationarity | $\epsilon$-stationarity (Moreau envelope) | DET | SO |
|:---:|:---:|:---:|:---:|:---:|
| SGD (Davis & Drusvyatskiy, 2019) | – | $\mathcal{O}\left(\frac{\rho}{\epsilon^4}\right)$ | ✓ | ✓ |
| SGDM (Mai & Johansson, 2020) | – | $\mathcal{O}\left(\frac{\rho^2}{\epsilon^4}\right)$ | ✓ | ✓ |
| INGD (Davis et al., 2022) | $\mathcal{O}\left(\frac{d\log(\rho)}{\delta\epsilon}\right)$ | – | ✗ | ✗ |
| D-O2NC with Option-I (ours) | $\mathcal{O}\left(\frac{1}{\delta\epsilon^3} + \rho^3\delta^2 + \frac{1}{\delta}\right)$ | – | ✓ | ✓ |
| D-O2NC with Option-II (ours) | $\mathcal{O}\left(\frac{1}{\delta\epsilon^3} + \frac{\rho^{7/3}\delta^{4/3}}{\epsilon^{3/2}} + \frac{1}{\delta}\right)$ | – | ✓ | ✓ |

Table 1: Comparison of subgradient-based methods for weakly convex optimization with respect to complexity bounds, determinism (DET), and applicability to stochastic oracle (SO). The notation used include convergence precisions $(\delta, \epsilon)$, weak convexity parameter $\rho$, and model dimension $d$.

## 4 EXPERIMENTS

In this section, we present a preliminary experimental study on the effectiveness of D-O2NC (configured with the periodically restarted OGD optimizer, Option-II) for training deep neural networks. Given that our algorithm corresponds to a momentum-restarted variant of SGDM, we adopt standard SGDM as the baseline for comparison. Additional experimental results are provided in Appendix E.

### 4.1 EXPERIMENT SETUP

**Dataset and backbone networks.** Our experiments are conducted on the CIFAR-10 image classification benchmark Krizhevsky & Hinton (2009) popularly used for evaluating deep learning models and algorithms. This dataset comprises 60,000 color images across 10 classes, with 50,000 images designated for training and 10,000 for testing. We adopt ResNet-101 (He et al., 2016) and Vision Transformer (ViT) (Dosovitskiy et al., 2021) as backbone networks for representation learning, with GeLU (Hendrycks & Gimpel, 2016) used as the activation function for both architectures. Notably, neural networks with smooth activation functions (e.g., GeLU, softplus) typically match or even outperform their ReLU-based non-smooth counterparts (Clevert et al., 2016; Xu et al., 2015).

**Implementation details and performance metrics.** For all considered algorithms, model parameters are optimized for 400 epochs with a mini-batch size of 256 for ResNet-101, and for 600 epochs (same mini-batch size) for ViT—where a patch size of 4 is used for the latter. The total number of mini-batches per epoch is 196. The initial learning rate is set to 0.01 and decayed via cosine annealing to enable smoother convergence. The optimizer employs a momentum of 0.99 and a weight decay of $5 \times 10^{-4}$. Our periodically restarted O2NC method is implemented and evaluated under two distinct restart frequencies: $T \in \{20, 50\}$. For each experiment, we carried out three independent runs with distinct random seeds, recording the empirical loss and training accuracy during training, as well as the test set prediction accuracy.

### 4.2 RESULTS

Figure 1 shows the convergence curves of the considered algorithms. Results for the optimal iterate of each trial are reported in Table 2. These results show that D-O2NC converges significantly faster than SGDM on both models: on average, D-O2NC outperforms SGDM in test accuracy by 0.9

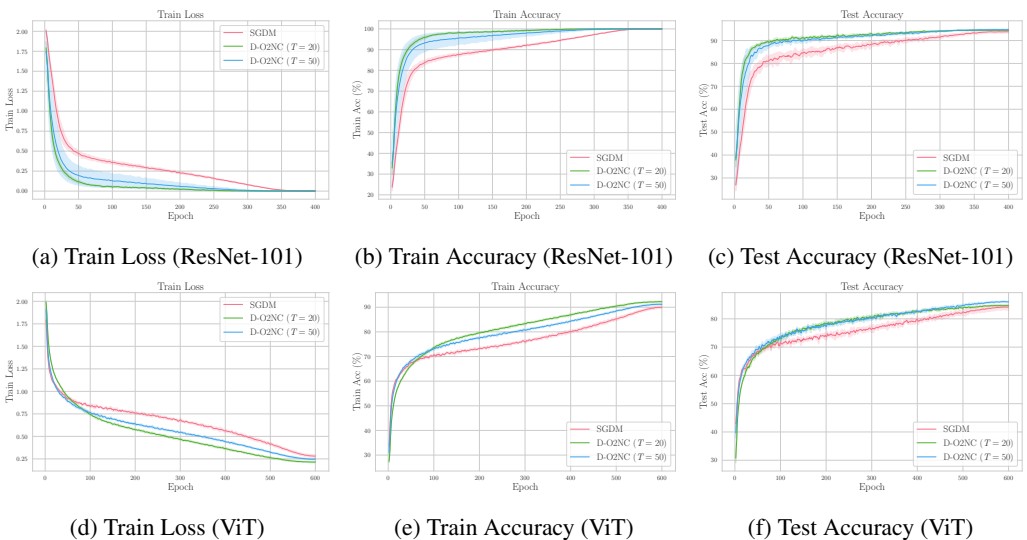

Figure 1: Experimental results on CIFAR-10 with ResNet-101 (top) and ViT (bottom) networks.

percentage points for ResNet-101 and 1.96 percentage points for ViT. These findings confirm that the momentum restart mechanism in our method not only accelerates convergence but also yields superior generalization performance.

Table 2: Numerical results of the optimal iterate per trial on CIFAR-10.

| Backbone | Metric | SGDM | D-O2NC $(T = 20)$ | D-O2NC $(T = 50)$ |
|---|---|---|---|---|
| ResNet-101 | Train Loss $(\times 10^{-3})$ | $2.55 \pm 0.08$ | $0.55 \pm 0.02$ | $1.38 \pm 0.01$ |
| | Train Accuracy $(\%)$ | $99.98 \pm 0.0$ | $100 \pm 0.0$ | $100 \pm 0.0$ |
| | Test Accuracy $(\%)$ | $93.91 \pm 0.55$ | $94.64 \pm 0.07$ | $94.81 \pm 0.34$ |
| ViT | Train Loss $(\times 10^{-1})$ | $2.78 \pm 0.10$ | $2.12 \pm 0.03$ | $2.44 \pm 0.10$ |
| | Train Accuracy $(\%)$ | $90.16 \pm 0.51$ | $92.38 \pm 0.28$ | $91.36 \pm 0.22$ |
| | Test Accuracy $(\%)$ | $84.26 \pm 0.41$ | $84.92 \pm 0.17$ | $86.22 \pm 0.29$ |

To validate the hyperparameter sensitivity of our method, we conduct additional experiments across diverse configurations of restart frequency, learning rate, and momentum coefficient (see Appendix E.1 for detailed results). These results demonstrate that D-O2NC consistently outperforms SGDM under identical hyperparameter configurations.

## 5 CONCLUSION

In this paper, we make progress towards resolving a critical issue regarding the link between O2NC (online-to-non-convex conversion) and SGDM: under auxiliary random interpolation or scaling, O2NC mirrors SGDM, but such randomization induces deviations from standard SGDM. To this end, for a broad class of weakly convex functions, we introduce D-O2NC as a derandomized version of O2NC that maintains the optimal oracle complexity $\mathcal{O}(\delta^{-1}\epsilon^{-3})$ while recovering SGDM in a deterministic manner. Our method allows the weak convexity parameter to scale up to $\mathcal{O}(\delta^{-1}\epsilon^{-1})$ without dominating the optimal rate, meaning that stronger stationarity enables tolerating higher non-convexity. Furthermore, we develop a periodically restarted variant of D-O2NC, which facilitates more progressive updates when far from stationarity. Corresponding to a momentum-restarted SGDM method, this variant has been empirically shown to be effective for training ResNet and ViT models on benchmark datasets. An interesting direction for future work is to extend our periodically restarted O2NC technique to the analysis and improvement of other popular machine learning optimizers, including Adam and schedule-free SGD.

ACKNOWLEDGMENTS

This work was supported by the Natural Science Foundation of China (NSFC) under Grant No. U21B2049, the Special Fund for Key Program of Science and Technology of Jiangsu Province under Grant No. BG2024042, the Gusu Leading Talents Program for Innovation and Entrepreneurship under Grant No. ZXL2025323, the CCF-Ant Research Fund (Ant Group) under Grant No. 20240512, and the "111 Center" under Grant No. B26023. We would like to thank the ICLR reviewers for their valuable feedback and suggestions.

DECLARATION OF LARGE LANGUAGE MODELS (LLMS) USAGE

We acknowledge the use of Doubao 1.6 (ByteDance, 2025) as an auxiliary LLM tool solely for polishing this manuscript, including refining English grammar and improving the readability of non-core descriptive content. All outputs from Doubao were thoroughly reviewed, revised, and validated by the authors to ensure accuracy, consistency with the research context, and alignment with academic standard. Doubao did not participate in idea formalization, theoretical derivation, result analysis, or drafting of core sections (Abstract, Introduction, Method, Experiment). All authors bear full responsibility for the manuscript's content integrity and scientific validity.

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

# A  PROOFS IN SECTION 2

## A.1  PROOF OF LEMMA 2

We prove the following restated lemma, which establishes the equivalence between $(\mu, \epsilon)$-regularized stationarity and Goldstein $(\delta, \epsilon)$-stationarity.

**Lemma 2.** *Let $\delta, \epsilon, \mu > 0$ be arbitrary positive values. Consider a Lipschitz function $f$.*

*(a) If $w$ is a $(\delta, \epsilon)$-stationary point, then it is also a $\left(\frac{\epsilon}{\delta^2}, 2\epsilon\right)$-regularized stationary point.*

*(b) If $w$ is a $(\mu, \epsilon)$-regularized stationary point, then it is also a $\left(\sqrt{\frac{\epsilon}{\mu}}, \epsilon\right)$-stationary point.*

*Proof.* Part(a): Let $w$ be a Goldstein $(\delta, \epsilon)$-stationary point of $f$, and set $\mu = \frac{\epsilon}{\delta^2}$. By the definition of the $\mu$-regularized subdifferential norm given in equation 2, we have

$$\|\partial f(w)\|_{+\mu} \leq \text{dist}(0, \partial_{\mathbb{B}_\delta(w)} f) + \mu \sup_{v \in \mathbb{B}_\delta(w)} \|v - w\|^2$$

$$= \text{dist}(0, \partial_\delta f(w)) + \mu \sup_{v \in \mathbb{B}_\delta(w)} \|v - w\|^2$$

$$\leq \epsilon + \mu \delta^2 = 2\epsilon,$$

where the second inequality uses the definition of Goldstein $(\delta, \epsilon)$-stationarity. It then follows from Definition 2 that $w$ is a $\left(\frac{\epsilon}{\delta^2}, 2\epsilon\right)$-regularized stationary point of $f$.

Part(b): We now consider the case where $w$ is a $(\mu, \epsilon)$-regularized stationary point of $f$. Let $\delta = \sqrt{\frac{\epsilon}{\mu}}$ and let $\varepsilon > 0$ be arbitrary. Since $\|\partial f(w)\|_{+\mu} \leq \epsilon$, Definition 2 ensures that there exists a set $V^*(\varepsilon)$ such that

$$\epsilon \geq \|\partial f(w)\|_{+\mu} \geq \text{dist}(0, \partial_{V^*(\varepsilon)} f) + \mu \sup_{v \in V^*(\varepsilon)} \|v - w\|^2 - \varepsilon,$$

which immediately yields

$$\text{dist}(0, \partial_{V^*(\varepsilon)} f) \leq \epsilon + \varepsilon, \quad \sup_{v \in V^*(\varepsilon)} \|v - w\| \leq \sqrt{\frac{\epsilon + \varepsilon}{\mu}} \leq \sqrt{\frac{\epsilon}{\mu}} + \sqrt{\frac{\varepsilon}{\mu}} = \delta + \sqrt{\frac{\varepsilon}{\mu}}.$$

The second inequality above implies that $V^*(\varepsilon) \subseteq \mathbb{B}_{\delta + \sqrt{\frac{\varepsilon}{\mu}}}(w)$ and hence $\partial_{V^*(\varepsilon)} f \subseteq \partial_{\delta + \sqrt{\frac{\varepsilon}{\mu}}} f$. We therefore obtain

$$\text{dist}\left(0, \partial_{\delta + \sqrt{\frac{\varepsilon}{\mu}}} f\right) \leq \text{dist}\left(0, \partial_{V^*(\varepsilon)} f\right) \leq \epsilon + \varepsilon.$$

Since $\varepsilon > 0$ can be chosen arbitrarily small and $\delta = \sqrt{\frac{\epsilon}{\mu}}$, it follows that $w$ is a Goldstein $\left(\sqrt{\frac{\epsilon}{\mu}}, \epsilon\right)$-stationary point. The proof is completed. $\qquad\square$

## A.2  PROOF OF LEMMA 3

Here we prove the following restated lemma on the monotonicity of $\|\partial f(w)\|_{+\mu}$ with respect to $\mu$.

**Lemma 3.** *Let $f$ be a Lipschitz function. Then for any $w \in \mathbb{R}^d$ and $0 < \mu_1 \leq \mu_2$, it holds that $\|\partial f(w)\|_{+\mu_1} \leq \|\partial f(w)\|_{+\mu_2}$.*

*Proof.* Fix a vector $w$ and let $\varepsilon > 0$ be arbitrary. By definition, there exists a subset $V_2^*(\varepsilon) \subseteq \mathbb{R}^d$ such that $\|\partial f(w)\|_{+\mu_2} \geq \text{dist}(0, \partial_{V_2^*(\varepsilon)} f) + \mu_2 \sup_{v \in V_2^*(\varepsilon)} \|v - w\|^2 - \varepsilon$. Using the definition again together with the condition $\mu_1 \leq \mu_2$, we obtain

$$\|\partial f(w)\|_{+\mu_1} \leq \text{dist}(0, \partial_{V_2^*(\varepsilon)} f) + \mu_1 \sup_{v \in V_2^*(\varepsilon)} \|v - w\|^2$$

$$\leq \text{dist}(0, \partial_{V_2^*(\varepsilon)} f) + \mu_2 \sup_{v \in V_2^*(\varepsilon)} \|v - w\|^2$$

$$\leq \|\partial f(w)\|_{+\mu_2} + \varepsilon.$$

Since $\varepsilon > 0$ can be chosen arbitrarily small, it follows that $\|\partial f(w)\|_{+\mu_1} \leq \|\partial f(w)\|_{+\mu_2}$. The proof is completed. $\qquad\square$

# B  PROOFS IN SECTION 3

## B.1  SOME KEY LEMMAS

The following lemma is crucial for analyzing Algorithm 2.

**Lemma 4.** *Suppose that Assumptions 1 and Assumption 2 hold. Let $\gamma \geq \rho$ and $D > 0$ be arbitrary numbers, and let $\eta \leq \frac{1}{8\gamma}$. Then for any $k \in [K]$, the sequence $\{w_t^{(k)}\}_{t=1}^T$ generated by Algorithm 2 satisfies*

$$\mathbb{E}\left[ R(w_T^{(k)}) - R(w_0^{(k)}) + \sum_{t=1}^T \frac{\gamma}{8}\|\Delta_t^{(k)}\|^2 \right]$$
$$\leq -\mathbb{E}\left[ DT \left\| \bar{g}^{(k)} \right\| \right] + \eta G^2 T + DG\sqrt{T} + \left( \gamma T + \frac{1}{\eta} \right) D^2 + \frac{\|\Delta_1^{(k)}\|^2}{\eta}.$$

*Proof.* Let us consider the filtration $\mathcal{F}_t = \sigma\{\Delta_1, \Delta_2, ..., \Delta_{t+1}\}$, where $\sigma\{\cdot\}$ denotes the generated $\sigma$-field. For any $n \geq 1$, it follows from Assumption 2 that $g_n := \mathbb{E}[\hat{g}_n \mid \mathcal{F}_{n-1}] \in \partial R(w_n)$. For any $\gamma \geq \rho$, the weak convexity condition in Assumption 1 implies that for all $n \geq 1$,

$$R(w_n) - R(w_{n-1}) \leq \langle g_n, \Delta_n \rangle + \frac{\gamma}{2}\|\Delta_n\|^2 = \mathbb{E}\left[ \langle \hat{g}_n, \Delta_n \rangle + \frac{\gamma}{2}\|\Delta_n\|^2 \mid \mathcal{F}_{n-1} \right].$$

By the law of total expectation,

$$\mathbb{E}\left[ R(w_n) - R(w_{n-1}) \right] \leq \mathbb{E}\left[ \langle \hat{g}_n, \Delta_n \rangle + \frac{\gamma}{2}\|\Delta_n\|^2 \right].$$

Fix an arbitrary $k \in [K]$. Recall that $w_t^{(k)} = w_{(k-1)T+t}$, $t \in [T]$ and we use analogous notation for $\Delta_t^{(k)}, \hat{g}_t^{(k)}, g_t^{(k)}$. Then the preceding inequality yields that for any $t \in [T]$:

$$\mathbb{E}\left[ R(w_t^{(k)}) - R(w_{t-1}^{(k)}) \right] \leq \mathbb{E}\left[ \langle \hat{g}_t^{(k)}, \Delta_t^{(k)} \rangle + \frac{\gamma}{2}\|\Delta_t^{(k)}\|^2 \right].$$

Summing the above bound over $t = 1, ..., T$ yields

$$\mathbb{E}\left[ R(w_T^{(k)}) - R(w_0^{(k)}) \right]$$
$$= \mathbb{E}\left[ \sum_{t=1}^T \left( R(w_t^{(k)}) - R(w_{t-1}^{(k)}) \right) \right] \leq \mathbb{E}\left[ \sum_{t=1}^T \left( \langle \hat{g}_t^{(k)}, \Delta_t^{(k)} \rangle + \frac{\gamma}{2}\|\Delta_t^{(k)}\|^2 \right) \right]. \tag{3}$$

We next upper-bound the right-hand side (RHS) in the above inequality using the OGD regret bound provided in Lemma 6. To this end, observe that the sequence $\{\Delta_t^{(k)}\}_{t=1}^T$ generated by Algorithm 2 corresponds to the iterates produced by OGD initialized at $\Delta_1^{(k)}$ with step-size $\eta$, applied to the sequence of quadratic losses $\{f_t^{(k)}\}_{t \in [T]}$ over the constraint $\mathbb{B}_D(0)$ (for Option-I) or over the full space $\mathbb{R}^d$ (for Option-II), where

$$f_t^{(k)}(\cdot) = \langle \hat{g}_t^{(k)}, \cdot \rangle + \frac{\gamma}{2}\| \cdot \|^2.$$

For any $D > 0$, we define the following comparator:

$$\bar{\Delta}^{(k)} := -D \frac{\sum_{t=1}^T g_t^{(k)}}{\left\| \sum_{t=1}^T g_t^{(k)} \right\|}.$$

Denote $\bar{g}^{(k)} := \frac{1}{T}\sum_{t=1}^{T} g_t^{(k)}$ and $\bar{\hat{g}}^{(k)} := \frac{1}{T}\sum_{t=1}^{T} \hat{g}_t^{(k)}$. Since $\eta \leq \frac{1}{8\gamma}$, we can apply Lemma 6 and obtain

$$
\mathbb{E}\left[\left(\sum_{t=1}^{T}\langle \hat{g}_t^{(k)}, \Delta_t^{(k)}\rangle + \frac{\gamma}{2}\|\Delta_t^{(k)}\|^2\right)\right]
$$

$$
\leq \mathbb{E}\left[\sum_{t=1}^{T}\left(\left\langle \hat{g}_t^{(k)}, \bar{\Delta}^{(k)}\right\rangle + \frac{\gamma}{2}\|\bar{\Delta}^{(k)}\|^2\right) + \sum_{t=1}^{T}\left(\eta\|\hat{g}_t^{(k)}\|^2 + \frac{\gamma}{2}\|\bar{\Delta}^{(k)}\|^2 - \frac{\gamma}{8}\|\Delta_t^{(k)}\|^2\right)\right.
$$

$$
\left. + \frac{1}{\eta}\left(\|\Delta_1^{(k)}\|^2 + \|\bar{\Delta}^{(k)}\|^2\right)\right]
$$

$$
= \mathbb{E}\left[\left\langle \sum_{t=1}^{T}(\hat{g}_t^{(k)} - g_t^{(k)}), \bar{\Delta}^{(k)}\right\rangle + \left\langle \sum_{t=1}^{T} g_t^{(k)}, \bar{\Delta}^{(k)}\right\rangle \right.
$$

$$
\left. + \sum_{t=1}^{T}\left(\eta\|\hat{g}_t^{(k)}\|^2 + \gamma\|\bar{\Delta}^{(k)}\|^2 - \frac{\gamma}{8}\|\Delta_t^{(k)}\|^2\right) + \frac{1}{\eta}\left(\|\Delta_1^{(k)}\|^2 + \|\bar{\Delta}^{(k)}\|^2\right)\right]
$$

$$
\overset{\zeta_1}{\leq} \mathbb{E}\left[DT\left\|\bar{\hat{g}}^{(k)} - \bar{g}^{(k)}\right\| - DT\left\|\bar{g}^{(k)}\right\| + \sum_{t=1}^{T}\left(\eta\|\hat{g}_t^{(k)}\|^2 + \gamma D^2 - \frac{\gamma}{8}\|\Delta_t^{(k)}\|^2\right) + \frac{D^2}{\eta} + \frac{\|\Delta_1^{(k)}\|^2}{\eta}\right]
$$

$$
\overset{\zeta_2}{\leq} -\mathbb{E}\left[DT\left\|\bar{g}^{(k)}\right\| + \sum_{t=1}^{T}\frac{\gamma}{8}\|\Delta_t^{(k)}\|^2\right] + \eta TG^2 + DG\sqrt{T} + \left(\gamma T + \frac{1}{\eta}\right)D^2 + \frac{\|\Delta_1^{(k)}\|^2}{\eta},
$$

where in step "$\zeta_1$" we used Cauchy–Schwarz inequality together with the fact that $\|\Delta_1^{(k)}\| \leq D$ holds for both Option-I and Option-II, and in step "$\zeta_2$" we used the bound $\|\hat{g}_t^{(k)}\| \leq G$ from Assumption 1 as well as the inequality $\mathbb{E}\left[\left\|\bar{\hat{g}}^{(k)} - \bar{g}^{(k)})\right\|\right] \leq \sqrt{\mathbb{E}\left[\left\|\bar{\hat{g}}^{(k)} - \bar{g}^{(k)}\right\|^2\right]} = \frac{1}{T}\sqrt{\sum_{t=1}^{T}\mathbb{E}\|g_t^{(k)} - \hat{g}_t^{(k)}\|^2} \leq \frac{1}{T}\sqrt{\sum_{t=1}^{T}\mathbb{E}\|\hat{g}_t^{(k)}\|^2} \leq \frac{G}{\sqrt{T}}$. Substituting the above inequality into equation 3 and rearranging terms yields the desired bound, which completes the proof. $\square$

The following simple lemma is also useful in our analysis. We provide its proof for the sake of completeness.

**Lemma 5.** *Let $w_1, w_2, ..., w_n$ be a set of vectors, and let $\bar{w} = \frac{1}{n}\sum_{i=1}^{n} w_i$. Then for all $i \in [n]$,*

$$
\|w_i - \bar{w}\|^2 \leq \frac{1}{n}\sum_{i'=1}^{n}\|w_i - w_{i'}\|^2 \leq n\sum_{j=1}^{n}\|\Delta_j\|^2,
$$

*where $\Delta_j := w_j - w_{j-1}$ and $w_0$ can be chosen arbitrarily.*

*Proof.* Fix any $i \in [n]$. We have

$$
\|w_i - \bar{w}\|^2 = \left\|w_i - \frac{1}{n}\sum_{i'=1}^{n} w_{i'}\right\|^2 \leq \frac{1}{n}\sum_{i'=1}^{n}\|w_i - w_{i'}\|^2
$$

$$
= \frac{1}{n}\sum_{i'=1}^{n}\left\|\sum_{j=i\wedge i'+1}^{i\vee i'}(w_{j-1} - w_j)\right\|^2
$$

$$
\leq \frac{1}{n}\sum_{i'=1}^{n}\left(\sum_{j=i\wedge i'+1}^{i\vee i'}\|\Delta_j\|\right)^2 \leq \left(\sum_{j=2}^{n}\|\Delta_j\|\right)^2 \leq n\sum_{j=1}^{n}\|\Delta_j\|^2.
$$

This completes the proof. $\square$

## B.2 PROOF OF THEOREM 1

**Theorem 1.** *Suppose that Assumption 1 and Assumption 2 hold. Let $\gamma \geq \rho$ be an arbitrary scalar, $\eta \leq \frac{1}{8\gamma}$, $K, T$ be positive integers, and $D$ be an arbitrary positive number. Then for any $\delta \geq TD$, the sequence $\{\bar{w}^{(k)}\}_{k=1}^{K}$ generated by Algorithm 2 with Option-I satisfies*

$$\mathbb{E}\left[\frac{1}{K}\sum_{k=1}^{K} dist(0, \partial_\delta R(\bar{w}^{(k)}))\right] \leq \frac{\eta G^2}{D} + \left(\gamma T + \frac{2}{\eta}\right)\frac{D}{T} + \frac{G}{\sqrt{T}} + \frac{\Delta R_0}{DKT}.$$

*Proof.* Under the given conditions, for any $k \in [K]$, we can invoke Lemma 4 on Algorithm 2 (with Option-I) to get

$$\mathbb{E}\left[R(w_T^{(k)}) - R(w_0^{(k)}) + \sum_{t=1}^{T}\frac{\gamma}{8}\|\Delta_t^{(k)}\|^2\right]$$

$$\leq -\mathbb{E}\left[DT\left\|\bar{g}^{(k)}\right\|\right] + \eta G^2 T + DG\sqrt{T} + \left(\gamma T + \frac{1}{\eta}\right)D^2 + \frac{\|\Delta_1^{(k)}\|^2}{\eta}$$

$$\leq -\mathbb{E}\left[DT\left\|\bar{g}^{(k)}\right\|\right] + \eta G^2 T + DG\sqrt{T} + \left(\gamma T + \frac{2}{\eta}\right)D^2,$$

where the last step uses the fact that $\|\Delta_1^{(k)}\| \leq D$, which follows from the explicit constraint imposed in Option-I. Note that by definition, $w_T^{(k)} = w_0^{(k+1)}$. Omitting the non-negative summation term on the LHS of the above inequality, we get

$$\mathbb{E}\left[R(w_0^{(k+1)}) - R(w_0^{(k)})\right] \leq -\mathbb{E}\left[DT\left\|\bar{g}^{(k)}\right\|\right] + \eta G^2 T + DG\sqrt{T} + \left(\gamma T + \frac{2}{\eta}\right)D^2.$$

Rearranging the terms on both sides of the above inequality yields

$$\mathbb{E}\left[DT\left\|\bar{g}^{(k)}\right\|\right] \leq \eta G^2 T + DG\sqrt{T} + \left(\gamma T + \frac{2}{\eta}\right)D^2 + \mathbb{E}\left[R(w_0^{(k)}) - R(w_0^{(k+1)})\right].$$

By summing the above inequality over $k \in [K]$ we get

$$\mathbb{E}\left[DT\sum_{k=1}^{K}\left\|\bar{g}^{(k)}\right\|\right] \leq \eta G^2 KT + DGK\sqrt{T} + \left(\gamma T + \frac{2}{\eta}\right)KD^2 + \mathbb{E}\left[\sum_{k=1}^{K}\left(R(w_0^{(k)}) - R(w_0^{(k+1)})\right)\right]$$

$$= \eta G^2 KT + DGK\sqrt{T} + \left(\gamma T + \frac{2}{\eta}\right)KD^2 + \mathbb{E}\left[R(w_0^{(1)}) - R(w_0^{(K+1)})\right]$$

$$\leq \eta G^2 KT + DGK\sqrt{T} + \left(\gamma T + \frac{2}{\eta}\right)KD^2 + R(w_0) - R^*.$$

Dividing both sides of the above inequality by $DKT$ yields

$$\mathbb{E}\left[\frac{1}{K}\sum_{k=1}^{K}\left\|\bar{g}^{(k)}\right\|\right] \leq \frac{\eta G^2}{D} + \left(\gamma T + \frac{2}{\eta}\right)\frac{D}{T} + \frac{G}{\sqrt{T}} + \frac{\Delta R_0}{DKT}. \tag{4}$$

Since $\|\Delta_t^{(k)}\| \leq D$ almost surely for all $t \in [T]$, applying Lemma 5 gives

$$\left\|w_t^{(k)} - \bar{w}^{(k)}\right\| \leq \sqrt{T\sum_{t=1}^{T}\|\Delta_t^{(k)}\|^2} \leq TD \leq \delta, \ \ \forall t \in [T],$$

which implies

$$dist\left(0, \partial_\delta R(\bar{w}^{(k)})\right) \leq \left\|\frac{1}{T}\sum_{t=1}^{T}g_t^{(k)}\right\| = \left\|\bar{g}^{(k)}\right\|.$$

Combining the above with equation 4, we obtain

$$\mathbb{E}\left[\frac{1}{K}\sum_{k=1}^{K}dist\left(0, \partial_\delta R(\bar{w}^{(k)})\right)\right] \leq \frac{\eta G^2}{D} + \left(\gamma T + \frac{2}{\eta}\right)\frac{D}{T} + \frac{G}{\sqrt{T}} + \frac{\Delta R_0}{DKT}.$$

This completes the proof. □

## B.3 Proof of Corollary 1

**Corollary 1.** *Suppose that Assumption 1 and Assumption 2 hold. Let $\delta, \epsilon > 0$ denote the target Goldstein stationarity parameters, and $N$ be the total iteration budget. We set*

$$T = \left\lceil (\delta N)^{2/3} \right\rceil, \quad K = \left\lfloor \frac{N}{T} \right\rfloor, \quad \gamma = \frac{N^{1/3}}{\delta^{2/3}}, \quad \eta = \frac{1}{8N}, \quad D = \frac{\delta^{1/3}}{N^{2/3}}.$$

*Suppose further that*

$$N \geq \frac{(G^2 + G + \Delta R_0 + 17)^3}{\delta \epsilon^3} + \rho^3 \delta^2 + \frac{1}{\delta}.$$

*Then the output $\bar{w}_T$ by Algorithm 2 with Option-I satisfies*

$$\mathbb{E}\left[dist\left(0, \partial_\delta R(\bar{w}_T)\right)\right] \leq \epsilon.$$

*Proof.* The given choice of the hyperparameters ensures that $TD \leq \delta$. Under the condition on $N$ we can verify that

$$\gamma \geq \rho, \quad \gamma\eta = \frac{1}{8(\delta N)^{2/3}} \leq \frac{1}{8}.$$

All conditions of Theorem 1 are thus satisfied in our setting, and applying the theorem yields

$$\mathbb{E}\left[\frac{1}{K}\sum_{k=1}^{K} dist\left(0, \partial_\delta R(\bar{w}^{(k)})\right)\right]$$

$$\leq \frac{\eta G^2}{D} + \left(\gamma T + \frac{2}{\eta}\right)\frac{D}{T} + \frac{G}{\sqrt{T}} + \frac{\Delta R_0}{DKT}$$

$$\leq \left(\frac{G^2}{8} + 1 + 16 + G + \Delta R_0\right)\frac{1}{(\delta N)^{1/3}}$$

$$\leq \left(G^2 + G + 17 + \Delta R_0\right)\frac{1}{(\delta N)^{1/3}} \leq \epsilon,$$

where the last inequality follows from the condition on $N$. The desired bound is obtained by noting that $\bar{w}_T \sim \text{Uniform}(\{\bar{w}^{(k)} : k \in [K]\})$. This completes the proof. □

## B.4 Proof of Theorem 2

**Theorem 2.** *Suppose that Assumption 1 and Assumption 2 hold. Let $\gamma \geq \rho$ be an arbitrary scalar, $\eta \leq \frac{1}{8\gamma}$, $K, T$ be positive integers, and $D$ be an arbitrary positive number. Then for any $\mu \leq \frac{\gamma}{8DT^2}$, the sequence $\{\bar{w}^{(k)}\}_{k=1}^{K}$ generated by Algorithm 2 with Option-II satisfies*

$$\mathbb{E}\left[\frac{1}{K}\sum_{k=1}^{K}\left\|\partial R(\bar{w}^{(k)})\right\|_{+\mu}\right] \leq \frac{\eta G^2}{D} + \left(\gamma T + \frac{1}{\eta}\right)\frac{D}{T} + \frac{G}{\sqrt{T}} + \frac{\Delta R_0}{DKT}.$$

*Proof.* Under the given conditions, for any $k \in [K]$, we can invoke Lemma 4 on Algorithm 2 (with Option-II) to get

$$\mathbb{E}\left[R(w_T^{(k)}) - R(w_0^{(k)}) + \sum_{t=1}^{T}\frac{\gamma}{8}\|\Delta_t^{(k)}\|^2\right]$$

$$\leq -\mathbb{E}\left[DT\left\|\bar{g}^{(k)}\right\|\right] + \eta G^2 T + DG\sqrt{T} + \left(\gamma T + \frac{1}{\eta}\right)D^2 + \frac{\|\Delta_1^{(k)}\|^2}{\eta}$$

$$\leq -\mathbb{E}\left[DT\left\|\bar{g}^{(k)}\right\|\right] + \eta G^2 T + DG\sqrt{T} + \left(\gamma T + \frac{1}{\eta}\right)D^2,$$

where the last inequality uses the fact that $\|\Delta_1^{(k)}\| = 0$, which follows from the periodic restarting step in Option-II of Algorithm 2. Recall that by definition, $w_T^{(k)} = w_0^{(k+1)}$. This implies that

$$\mathbb{E}\left[ R(w_0^{(k+1)}) - R(w_0^{(k)}) + \underbrace{\frac{\gamma}{8} \sum_{t=1}^{T} \|\Delta_t^{(k)}\|^2}_{A} \right]$$
$$\leq - \mathbb{E}\left[ DT \left\| \bar{g}^{(k)} \right\| \right] + \eta G^2 T + DG\sqrt{T} + \left( \gamma T + \frac{1}{\eta} \right) D^2.$$

By applying Lemma 5, we can lower-bound the term $A^{(k)}$ on the left-hand side (LHS) of the above inequality as

$$A \geq \frac{\gamma}{8T} \max_{t \in [T]} \left\| w_t^{(k)} - \bar{w}^{(k)} \right\|^2.$$

It follows that

$$\mathbb{E}\left[ R(w_0^{(k+1)}) - R(w_0^{(k)}) + \frac{\gamma}{8T} \max_{t \in [T]} \left\| w_t^{(k)} - \bar{w}^{(k)} \right\|^2 \right]$$
$$\leq - \mathbb{E}\left[ DT \left\| \bar{g}^{(k)} \right\| \right] + \eta G^2 T + DG\sqrt{T} + \left( \frac{\gamma T}{2} + \frac{1}{\eta} \right) D^2.$$

Rearranging the terms on both sides of the above inequality yields

$$\mathbb{E}\left[ DT \left\| \bar{g}^{(k)} \right\| + \frac{\gamma}{8T} \max_{t \in [T]} \left\| w_t^{(k)} - \bar{w}^{(k)} \right\|^2 \right]$$
$$\leq \eta G^2 T + DG\sqrt{T} + \left( \frac{\gamma T}{2} + \frac{1}{\eta} \right) D^2 + \mathbb{E}\left[ R(w_0^{(k)}) - R(w_0^{(k+1)}) \right].$$

By summing the above inequality over $k \in [K]$, we get

$$\mathbb{E}\left[ DT \sum_{k=1}^{K} \left\| \bar{g}^{(k)} \right\| + \frac{\gamma}{8T} \sum_{k=1}^{K} \max_{t \in [T]} \left\| w_t^{(k)} - \bar{w}^{(k)} \right\|^2 \right]$$
$$\leq \eta G^2 KT + DGK\sqrt{T} + \left( \gamma T + \frac{1}{\eta} \right) KD^2 + \mathbb{E}\left[ \sum_{k=1}^{K} \left( R(w_0^{(k)}) - R(w_0^{(k+1)}) \right) \right]$$
$$= \eta G^2 KT + DGK\sqrt{T} + \left( \gamma T + \frac{1}{\eta} \right) KD^2 + \mathbb{E}\left[ R(w_0^{(1)}) - R(w_0^{(K+1)}) \right]$$
$$\leq \eta G^2 KT + DGK\sqrt{T} + \left( \gamma T + \frac{1}{\eta} \right) KD^2 + R(w_0) - R^*.$$

Finally, dividing both sides of the above inequality by $DKT$ yields

$$\mathbb{E}\left[ \frac{1}{K} \sum_{k=1}^{K} \left( \left\| \bar{g}^{(k)} \right\| + \frac{\gamma}{8DT^2} \max_{t \in [T]} \left\| w_t^{(k)} - \bar{w}^{(k)} \right\|^2 \right) \right] \leq \frac{\eta G^2}{D} + \left( \gamma T + \frac{1}{\eta} \right) \frac{D}{T} + \frac{G}{\sqrt{T}} + \frac{\Delta R_0}{DKT}.$$

Since $\mu \leq \mu' = \frac{\gamma}{8DT^2}$, by Lemma 3 we have

$$\left\| \partial R(\bar{w}^{(k)}) \right\|_{+\mu} \leq \left\| \partial R(\bar{w}^{(k)}) \right\|_{+\mu'} \leq \left\| \bar{g}^{(k)} \right\| + \frac{\gamma}{8DT^2} \max_{t \in [T]} \left\| w_t^{(k)} - \bar{w}^{(k)} \right\|^2.$$

Combining the preceding two inequalities leads to the desired result. This completes the proof. □

## B.5 PROOF OF COROLLARY 2

**Corollary 2.** *Suppose that Assumption 1 and Assumption 2 hold. Let $\mu, \epsilon > 0$ denote the target regularized stationarity parameters and $N$ be the total iteration budget. We set*

$$T = \left\lceil N^{4/7} \mu^{-2/7} \right\rceil, \quad K = \left\lfloor \frac{N}{T} \right\rfloor, \quad \gamma = N^{3/7} \mu^{2/7}, \quad \eta = \frac{1}{8N}.$$

*Suppose further that*

$$N \geq \frac{(4G^2 + 32\Delta R_0 + 1)^{7/2} \mu^{1/2}}{\epsilon^{7/2}} + \frac{\rho^{7/3}}{\mu^{2/3}} + \mu^{1/2}.$$

*Then the output $\bar{w}_T$ of Algorithm 2 with Option-II satisfies*

$$\mathbb{E}\left[\|\partial R(\bar{w}_T)\|_{+\mu}\right] \leq \epsilon.$$

*Proof.* Under the condition on $N$, we can verify that

$$\gamma \geq \rho, \quad \eta\gamma = \frac{\mu^{2/7}}{8N^{4/7}} \leq \frac{1}{8}.$$

We now consider the value $D = \frac{1}{32}\mu^{-1/7}N^{-5/7}$. Again the condition on $N$ implies that

$$T' := N^{4/7}\mu^{-2/7} \geq 1.$$

With the given choices of $T, \gamma, D$, it can be readily shown that

$$\frac{\gamma}{8DT^2} = \frac{\gamma}{8D\lceil T'\rceil^2} \geq \frac{\gamma}{8D(T'+1)^2} \geq \frac{\gamma}{32DT'^2} = \mu.$$

In view of the above arguments, all conditions of Theorem 2 are satisfied in our setting, and we can thus apply the theorem to obtain

$$\mathbb{E}\left[\frac{1}{K}\sum_{k=1}^{K}\left\|\partial R(\bar{w}^{(k)})\right\|_{+\mu}\right] \leq \frac{\eta G^2}{D} + \left(\gamma T + \frac{1}{\eta}\right)\frac{D}{T} + \frac{G}{\sqrt{T}} + \frac{\Delta R_0}{DKT}$$

$$\leq \left(4G^2 + \frac{1}{32} + \frac{1}{4} + 32\Delta R_0\right)\frac{\mu^{1/7}}{N^{2/7}}$$

$$\leq (4G^2 + 1 + 32\Delta R_0)\frac{\mu^{1/7}}{N^{2/7}} \leq \epsilon,$$

where the last inequality follows from the condition on $N$. The desired bound is obtained by noting that $\bar{w}_T \sim \text{Uniform}(\{\bar{w}^{(k)} : k \in [K]\})$. This completes the proof. $\qquad\square$

## C ANALYSIS OF ONLINE GRADIENT DESCENT FOR QUADRATIC LOSSES

Consider the quadratic loss functions of the form $f_t(x) = \langle u_t, x\rangle + \frac{\gamma}{2}\|x\|^2, t \geq 1$, defined over a convex constraint $\mathcal{C}$. We analyze the following standard online gradient descent (OGD) method, which starts from an initial iterate $x_1$ with step-size $\eta > 0$:

$$x_{t+1} := \Pi_{\mathcal{C}}\left[x_t - \eta\nabla f_t(x_t)\right] = \Pi_{\mathcal{C}}\left[(1 - \eta\gamma)x_t - \eta u_t\right], \qquad (5)$$

where $\Pi_{\mathcal{C}}$ denotes the Euclidian projection operator associated with $\mathcal{C}$. Let $\text{Regret}_T(\bar{x})$ denote the regret of the algorithm with respect to some comparator $\bar{x} \in \mathcal{C}$ after $T$ iterations, defined as follows:

$$\text{Regret}_T(\bar{x}) := \sum_{t=1}^{T} f_t(x_t) - \sum_{t=1}^{T} f_t(\bar{x}).$$

Based on standard analysis, we can establish the following result regarding the regret bound of the OGD algorithm described above.

**Lemma 6.** *Suppose that $\eta\gamma \leq \frac{1}{8}$. Then the OGD procedure 5 applied to $\{f_t\}_{t=1}^{T}$ over the convex constraint set $\mathcal{C}$ guarantees that for all $T \geq 1$ and all $\bar{x} \in \mathcal{C}$:*

$$\text{Regret}_T(\bar{x}) \leq \sum_{t=1}^{T}\left(\eta\|u_t\|^2 + \frac{\gamma}{2}\|\bar{x}\|^2 - \frac{\gamma}{8}\|x_t\|^2\right) + \frac{1}{\eta}\left(\|x_1\|^2 + \|\bar{x}\|^2\right).$$

*Proof.* First, it can be verified that

$$\|x_{t+1} - \bar{x}\|^2 = \|\Pi_{\mathcal{C}}\left(x_t - \eta \nabla f_t(x_t)\right) - \bar{x}\|^2$$
$$\leq \|x_t - \eta \nabla f_t(x_t) - \bar{x}\|^2$$
$$= \|x_t - \bar{x}\|^2 + \eta^2 \|\nabla f_t(x_t)\|^2 - 2\eta \langle \nabla f_t(x_t), x_t - \bar{x}\rangle,$$

which implies

$$\langle \nabla f_t(x_t), x_t - \bar{x}\rangle \leq \frac{\|x_t - \bar{x}\|^2 - \|x_{t+1} - \bar{x}\|^2}{2\eta} + \frac{\eta \|\nabla f_t(x_t)\|^2}{2}.$$

Using the strong convexity of $f_t$, we can then show that

$$\text{Regret}_T(\bar{x}) = \sum_{t=1}^{T} f_t(x_t) - \sum_{t=1}^{T} f_t(\bar{x})$$

$$\leq \sum_{t=1}^{T} \langle \nabla f_t(x_t), x_t - \bar{x}\rangle - \frac{\gamma}{2}\|x_t - \bar{x}\|^2$$

$$\leq \sum_{t=1}^{T} \left(\frac{\|x_t - \bar{x}\|^2 - \|x_{t+1} - \bar{x}\|^2}{2\eta} - \frac{\gamma}{2}\|x_t - \bar{x}\|^2\right) + \sum_{t=1}^{T} \frac{\eta \|\nabla f_t(x_t)\|^2}{2}$$

$$= -\sum_{t=1}^{T} \frac{\gamma}{2}\|x_t - \bar{x}\|^2 + \frac{1}{2\eta}\|x_1 - \bar{x}\|^2 - \frac{1}{2\eta}\|x_{T+1} - \bar{x}\|^2 + \sum_{t=1}^{T} \frac{\eta \|u_t + \gamma x_t\|^2}{2}$$

$$\leq -\sum_{t=1}^{T} \frac{\gamma}{2}\|x_t - \bar{x}\|^2 + \frac{1}{\eta}\left(\|x_1\|^2 + \|\bar{x}\|^2\right) + \sum_{t=1}^{T} \eta(\|u_t\|^2 + \gamma^2\|x_t\|^2)$$

$$\overset{\zeta_1}{\leq} -\sum_{t=1}^{T} \frac{\gamma}{2}\left(\frac{\|x_t\|^2}{2} - \|\bar{x}\|^2\right) + \frac{1}{\eta}\left(\|x_1\|^2 + \|\bar{x}\|^2\right) + \sum_{t=1}^{T} \eta(\|u_t\|^2 + \gamma^2\|x_t\|^2)$$

$$= \sum_{t=1}^{T} \left(\eta\|u_t\|^2 + \frac{\gamma}{2}\|\bar{x}\|^2 - \gamma\left(\frac{1}{4} - \eta\gamma\right)\|x_t\|^2\right) + \frac{1}{\eta}\left(\|x_1\|^2 + \|\bar{x}\|^2\right)$$

$$\leq \sum_{t=1}^{T} \left(\eta\|u_t\|^2 + \frac{\gamma}{2}\|\bar{x}\|^2 - \frac{\gamma}{8}\|x_t\|^2\right) + \frac{1}{\eta}\left(\|x_1\|^2 + \|\bar{x}\|^2\right),$$

where step "$\zeta_1$" uses the fact $\|a - b\|^2 \geq \frac{\|a\|^2}{2} - \|b\|^2$, and the last step follows from the condition $\eta\gamma \leq \frac{1}{8}$. This completes the proof. $\square$

**Remark 9.** *The main message conveyed by Lemma 6 is that it is beneficial to control the scales of the competitor $\bar{x}$ and the initial iterate $x_1$ to keep the regret small, even when the domain of interest is allowed to be unbounded. This result inspires us to explicitly control the scale of the initial iterate.*

## D    FROM GOLDSTEIN TO CLARKE STATIONARITY

As a side contribution of our work, we establish in the following theorem a set of results regarding the connection between the Goldstein stationarity of a weakly convex function and the Clarke stationarity of its Moreau envelope, which we believe are of independent interest.

**Theorem 3.** *Let $f$ be a G-Lipschitz and $\rho$-weakly convex function.*

*(a) If $w$ is a $(\delta, \epsilon)$-stationary point of $f$, then it holds that*

$$\left\|\nabla f_{1/(3\rho)}(w)\right\| \leq 3\sqrt{\frac{\epsilon^2}{2} + 4G\rho\delta + 2\rho^2\delta^2}.$$

*(b) If $w$ is a $(\mu, \epsilon)$-regularized stationary point of $f$, then it holds that*

$$\left\|\nabla f_{1/(3\rho)}(w)\right\| \leq 3\sqrt{\frac{\epsilon^2}{2} + 4G\rho\sqrt{\frac{\epsilon}{\mu}} + 2\rho^2\frac{\epsilon}{\mu}}.$$

*Proof.* Part (a): Let $w$ be a $(\delta, \epsilon)$-stationary point of $f$. By definition, there exists a subset $V \subseteq \mathbb{B}_\delta(w)$ and a set of weights $\{\alpha_v\}_{v \in V}$ such that $\alpha_v \geq 0, \sum_{v \in V} \alpha_v = 1$, and

$$\left\| \sum_{v \in V} \alpha_v g_v \right\| \leq \epsilon, \tag{6}$$

where $g_v \in \partial f(v)$. For any $w'$, consider a subgradient $g' \in \partial f(w')$. Since $f$ is $\rho$-weakly convex, we can show that

$$f(w') = \sum_{v \in V} \alpha_v f(w')$$

$$\geq \sum_{v \in V} \alpha_v \left( f(v) + \langle g_v, w' - v \rangle - \frac{\rho}{2} \|w' - v\|^2 \right)$$

$$= f(w) + \left\langle \sum_{v \in V} \alpha_v g_v, w' - w \right\rangle + \sum_{v \in V} \alpha_v \left( f(v) - f(w) + \langle g_v, w - v \rangle - \frac{\rho}{2} \|w' - w + w - v\|^2 \right)$$

$$\overset{\zeta_1}{\geq} f(w) + \left\langle \sum_{v \in V} \alpha_v g_v, w' - w \right\rangle - \rho\|w' - w\|^2 + \sum_{v \in V} \alpha_v \left( f(v) - f(w) + \langle g_v, w - v \rangle - \rho\|w - v\|^2 \right)$$

$$\overset{\zeta_2}{\geq} f(w) - \frac{1}{4\rho} \left\| \sum_{v \in V} \alpha_v g_v \right\|^2 - 2\rho\|w' - w\|^2 - \sum_{v \in V} \alpha_v \left( 2G\|v - w\| + \rho\|w - v\|^2 \right)$$

$$\overset{\zeta_3}{\geq} f(w) - 2\rho\|w' - w\|^2 - \frac{\epsilon^2}{4\rho} - 2G\delta - \rho\delta^2$$

$$\geq f(w') + \langle g', w - w' \rangle - \frac{\rho}{2}\|w - w'\|^2 - 2\rho\|w' - w\|^2 - \frac{\epsilon^2}{4\rho} - 2G\delta - \rho\delta^2$$

$$= f(w') + \langle g', w - w' \rangle - \frac{5\rho}{2}\|w - w'\|^2 - \frac{\epsilon^2}{4\rho} - 2G\delta - \rho\delta^2,$$

where step "$\zeta_1$" uses the Cauchy–Schwarz inequality, step "$\zeta_2$" uses the Cauchy–Schwarz inequality and the $G$-Lipschitz continuity of $f$, step "$\zeta_3$" uses the fact that $V \subseteq \mathbb{B}_\delta(w)$ and equation 6, and the last inequality uses the $\rho$-weak convexity of $f$. We now consider $\hat{w} = \text{prox}_{f/\bar{\rho}}(w)$ for some $\bar{\rho} > \frac{5\rho}{2}$, which by Lemma 1 satisfies

$$\nabla f_{1/\bar{\rho}}(w) = \bar{\rho}(w - \hat{w}) \in \partial f(\hat{w}).$$

Subsisting $w' = \hat{w}$ into the preceding inequality and rearranging the terms yields

$$\|w - \hat{w}\|^2 \leq \left( \bar{\rho} - \frac{5\rho}{2} \right)^{-1} \left( \frac{\epsilon^2}{4\rho} + 2G\delta + \rho\delta^2 \right).$$

It follows from the above inequality that

$$\left\| \nabla f_{1/\bar{\rho}}(w) \right\| = \|\bar{\rho}(w - \hat{w})\| \leq \bar{\rho} \left( \bar{\rho} - \frac{5\rho}{2} \right)^{-1/2} \left( \frac{\epsilon^2}{4\rho} + 2G\delta + \rho\delta^2 \right)^{1/2}.$$

Finally, setting $\bar{\rho} = 3\rho$ in the above expression and performing some minor algebraic manipulations yields the desired bound in Part (a). The bound in Part (b) follows directly from Part (a) and Lemma 2. This completes the proof. $\qquad\square$

**Remark 10.** *Theorem 3 essentially shows that the $(\delta, \epsilon)$-stationarity of a weakly convex function implies the $(\epsilon + \sqrt{\delta})$-stationarity of its Moreau envelope, and correspondingly the $(\mu, \epsilon)$-regularized stationary implies the $(\epsilon + \sqrt{\epsilon/\mu})$-stationarity.*

**Remark 11.** *Conversely, for a $\rho$-weakly convex function $f$, the translation from the Clarke stationarity of its Moreau envelope to the Goldstein stationarity of the original objective is relatively straightforward. Indeed, suppose that $w$ is an $\epsilon$-stationary point of the Moreau envelope $f_{1/(2\rho)}$ such that $\|\nabla f_{1/(2\rho)}(w)\| \leq \epsilon$. Consider $\hat{w} := \text{prox}_{f/(2\rho)}(w)$. Then according to Lemma 1 we have*

$$\nabla f_{1/(2\rho)}(w) \in \partial f(\hat{w}), \quad \|w - \hat{w}\| \leq \frac{\|\nabla f_{1/(2\rho)}(w)\|}{2\rho} \leq \frac{\epsilon}{2\rho},$$

which implies $\text{dist}\left(0, \partial_{\frac{\epsilon}{2\rho}} f(w)\right) \leq \|\nabla f_{1/(2\rho)}(w)\| \leq \epsilon$, and thus $w$ is a $(\delta, \epsilon)$-stationary point of $f$ with $\delta = \frac{\epsilon}{2\rho}$. However, a limitation of translating rates via the setting $\delta = \epsilon/(2\rho)$ is that it excludes the range of relatively large $\delta$ (e.g., $\delta = \sqrt{\epsilon}$, a choice critical for sharper rates in second-order smooth functions), as $\rho$ is typically lower bounded by a constant.

The following corollary is a direct consequence of Theorem 3 when applied to Algorithm 2 with Option-I.

**Corollary 3.** *Suppose that Assumption 1 and Assumption 2 hold. Let $\epsilon > 0$ be the desired stationarity precision, and let $N$ be the total budget of iteration. Set*

$$T = \left\lceil \epsilon^{4/3} N^{2/3} \right\rceil, \quad K = \left\lfloor \frac{N}{T} \right\rfloor, \quad \gamma = \frac{N^{1/3}}{\epsilon^{4/3}}, \quad \eta = \frac{1}{8N}, \quad D = \frac{\epsilon^{2/3}}{N^{2/3}}.$$

*Suppose further that*

$$N \geq \frac{(G^2 + G + \Delta R_0 + 17)^3}{\epsilon^5} + \rho^3 \epsilon^4 + \frac{1}{\epsilon^2}.$$

*Then the output $\bar{w}_T$ of Algorithm 2 with Option-I satisfies*

$$\mathbb{E}\left[\left\|\nabla f_{1/(3\rho)}(\bar{w}_T)\right\|\right] \leq \mathcal{O}\left(\sqrt{G\rho}\epsilon + \rho\epsilon^2\right).$$

*Proof.* Let $\delta = \epsilon^2$ and $\varepsilon(\delta, \bar{w}_T) := \text{dist}(0, \partial_\delta R(\bar{w}_T))$. Under the given conditions, it follows from Corollary 1 that

$$\mathbb{E}\left[\varepsilon(\delta, \bar{w}_T)\right] = \mathbb{E}\left[\text{dist}(0, \partial_\delta R(\bar{w}_T))\right] \leq \epsilon. \tag{7}$$

Conditioned on $\bar{w}_T$, it is straightforward to see that $\bar{w}_T$ is a $(\delta, \varepsilon(\delta, \bar{w}_T))$-stationary point of $R$. Therefore, from Part (a) of Theorem 3 we have

$$\left\|\nabla R_{1/(3\rho)}(\bar{w}_T)\right\| \leq 3\sqrt{\frac{\varepsilon^2(\delta, \bar{w}_T)}{2} + 4G\rho\delta + 2\rho^2\delta^2} \leq \frac{3\sqrt{2}}{2}\varepsilon(\delta, \bar{w}_T) + 6\sqrt{G\rho\delta} + 3\sqrt{2}\rho\delta.$$

Taking expectation on both sides of the above inequality yields

$$\mathbb{E}\left[\left\|\nabla R_{1/(3\rho)}(\bar{w}_T)\right\|\right] \leq \mathbb{E}\left[\frac{3\sqrt{2}}{2}\varepsilon(\delta, \bar{w}_T) + 6\sqrt{G\rho\delta} + 3\sqrt{2}\rho\delta\right]$$

$$\leq \frac{3\sqrt{2}}{2}\epsilon + 6\sqrt{G\rho}\epsilon + 3\sqrt{2}\rho\epsilon^2,$$

where the last step uses equation 7 and $\delta = \epsilon^2$. This completes the proof. $\qquad\square$

**Remark 12.** *The $\mathcal{O}(\epsilon^{-5})$ complexity established in Corollary 3 is suboptimal compared to the optimal $\mathcal{O}(\epsilon^{-4})$ complexity of SGD (Davis & Grimmer, 2019) and SGDM (Mai & Johansson, 2020) for achieving the $\epsilon$-stationarity of the Moreau envelope. Such a slower rate is mainly attributed to the $\sqrt{\delta}$ component appearing in the bound of Theorem 3 (Part (a)), which remains an open problem for improvement in future work.*

Similarly, by applying Theorem 3 to Algorithm 2 with Option-II, we obtain the following corollary.

**Corollary 4.** *Suppose that Assumption 1 and Assumption 2 hold. Let $\epsilon > 0$ be the desired stationarity precision, and let $N$ be the total budget of iteration. Set*

$$T = \left\lceil N^{4/7} \epsilon^{6/7} \right\rceil, \quad K = \left\lfloor \frac{N}{T} \right\rfloor, \quad \gamma = N^{3/7} \epsilon^{-6/7}, \quad \eta = \frac{1}{8N}.$$

*Suppose further that*

$$N \geq \frac{(4G^2 + 1 + 32\Delta R_0)^{7/2}}{\epsilon^5} + \rho^{7/3}\epsilon^2 + \frac{1}{\epsilon^{3/2}}.$$

*Then the output $\bar{w}_T$ of Algorithm 2 with Option-II satisfies*

$$\mathbb{E}\left[\left\|\nabla f_{1/(3\rho)}(\bar{w}_T)\right\|\right] \leq \mathcal{O}\left(\sqrt{G\rho}\epsilon + \rho\epsilon^2\right).$$

*Proof.* The proof follows similarly to that of Corollary 3 and is included here for completeness. Let $\mu = \epsilon^{-3}$ and $\varepsilon(\mu, \bar{w}_T) := \|\partial R(\bar{w}_T)\|_{+\mu}$. Under the stated assumptions, it follows from Corollary 2 that

$$\mathbb{E}\left[\varepsilon(\mu, \bar{w}_T)\right] = \mathbb{E}\left[\|\partial R(\bar{w}_T)\|_{+\mu}\right] \leq \epsilon. \tag{8}$$

Conditioned on $\bar{w}_T$, it is straightforward to see that $\bar{w}_T$ is a $(\delta, \varepsilon(\delta, \bar{w}_T))$-stationary point of $R$. Accordingly, by Part (b) of Theorem 3, we obtain

$$\left\|\nabla R_{1/(3\rho)}(\bar{w}_T)\right\| \leq 3\sqrt{\frac{\varepsilon^2(\mu, \bar{w}_T)}{2} + 4G\rho\sqrt{\frac{\epsilon}{\mu}} + 2\rho^2\frac{\epsilon}{\mu}} \leq \frac{3\sqrt{2}}{2}\varepsilon(\mu, \bar{w}_T) + 6\sqrt{G\rho\sqrt{\frac{\epsilon}{\mu}}} + 3\sqrt{2}\rho\sqrt{\frac{\epsilon}{\mu}}.$$

Taking expectation on both sides of the above inequality yields

$$\mathbb{E}\left[\left\|\nabla R_{1/(3\rho)}(\bar{w}_T)\right\|\right] \leq \mathbb{E}\left[\frac{3\sqrt{2}}{2}\varepsilon(\mu, \bar{w}_T) + 6\sqrt{G\rho\sqrt{\frac{\epsilon}{\mu}}} + 3\sqrt{2}\rho\sqrt{\frac{\epsilon}{\mu}}\right]$$

$$\leq \frac{3\sqrt{2}}{2}\epsilon + 6\sqrt{G\rho}\epsilon + 3\sqrt{2}\rho\epsilon^2,$$

where the last inequality uses equation 8 and the choice $\mu = \epsilon^{-3}$. This establishes the desired bound. $\qquad\square$

# E ADDITIONAL EXPERIMENTAL RESULTS

In this appendix, we present additional experimental results on neural networks (Appendix E.1) and robust phase retrieval (Appendix E.2) to further validate the effectiveness and efficiency of our D-O2NC method equipped with periodically restarted OGD.

## E.1 EXPERIMENTS ON NEURAL NETWORKS

**Additional descriptions of backbone networks.** We employ the ResNet-101 and ViT models to evaluate our method. ResNet-101 is a representative deep architecture in the ResNet family, consisting of 101 layers constructed by stacking residual blocks, each formed by 1×1, 3×3, and 1×1 convolutional layers. It is widely used as a backbone in various downstream computer vision tasks, such as object detection and image segmentation. In our experiments, the employed ViT contains 6 Transformer encoder layers, each with 8 multi-head self-attention heads and a 512-dimensional multilayer perceptron (MLP). The dropout rate is set to 0.1, and the input is divided into 4 patches. Both networks are trained from scratch.

**Results under various restarting frequency.** In our experiments on the CIFAR-10 dataset, we set the restarting frequency $T$ to the range $\{2, 20, 50, 196\}$, and the total number of minibatches in one epoch is 196. Consistent with the parameter settings in the main paper, we use a learning rate of 0.01 and momentum of 0.99. As shown in Figure 2, the results demonstrate that, in most cases, the model performance first improves gradually with increasing $T$ and then degrades. Notably, extreme values of $T$ (e.g., $T = 2$) lead to degraded performance. Thus, choosing an appropriate restarting interval $T$ is crucial for achieving optimal model performance.

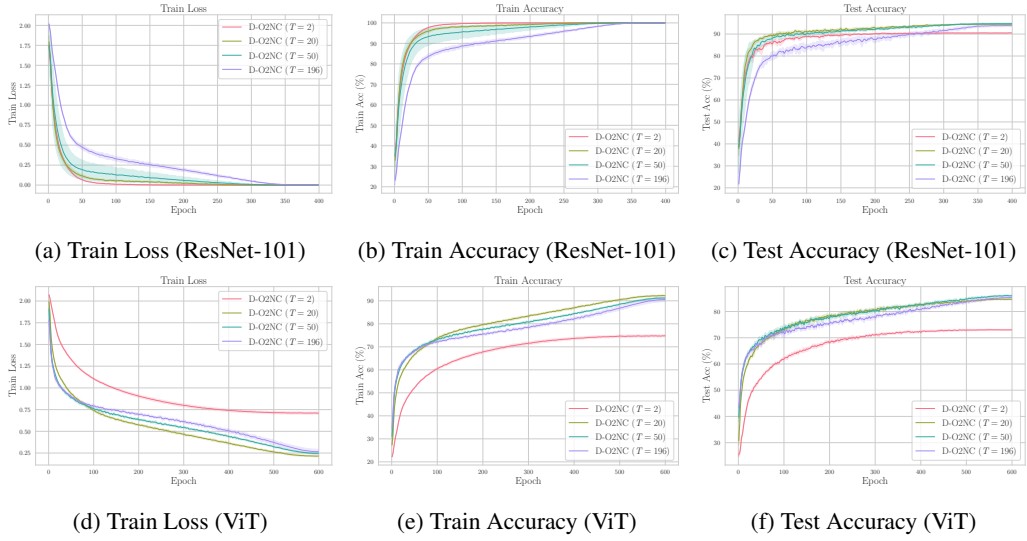

Figure 2: Experimental results on CIFAR-10 with ResNet-101 (top) and ViT (bottom) networks under various restarting frequencies $T$.

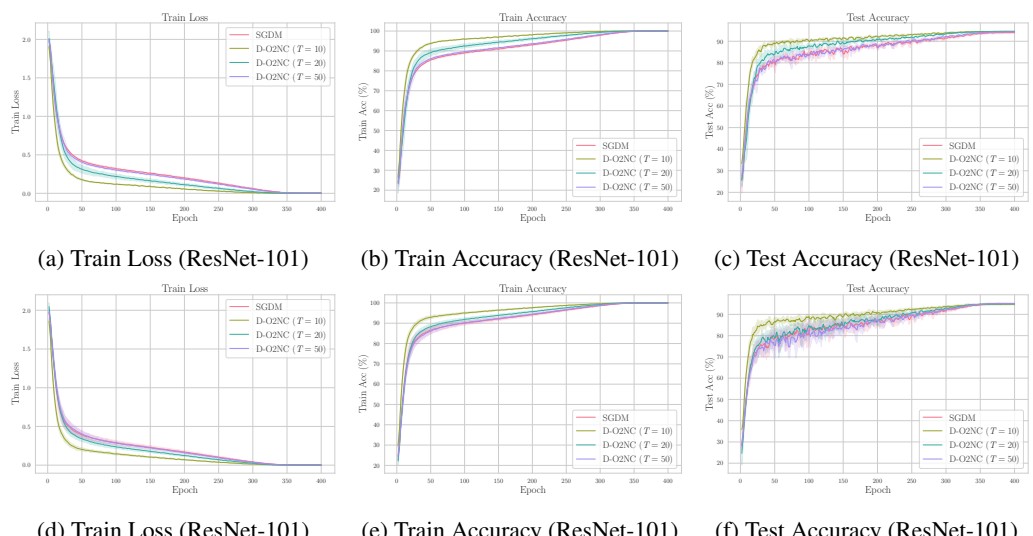

Figure 3: Experimental results on CIFAR-10 with ResNet-101 under relaxed momentum and learning rate settings. The top row corresponds to a momentum coefficient of 0.95 and learning rate of 0.05, while the bottom row corresponds to a momentum coefficient of 0.9 and learning rate of 0.1.

**Results under various momentum and learning rate configurations.** For the experiments reported in the main paper, we used a momentum coefficient of 0.99 and a learning rate of 0.01. In this section, we further present results under two additional parameter settings: $(0.95, 0.05)$ and $(0.9, 0.1)$. As shown in Figure 3, our method consistently achieves substantial advantages in both convergence speed and prediction accuracy. Moreover, it can be observed that the performance gain of our method gradually diminishes as $T$ increases under these hyperparameter settings.

**Results on CIFAR-100.** Finally, in addition to CIFAR-10, we also evaluate our algorithm on the CIFAR-100 dataset. CIFAR-100 is an advanced counterpart of CIFAR-10, comprising 60,000 $32{\times}32$ color images. In contrast to the 10 coarse classes in CIFAR-10, CIFAR-100 contains 100 fine-grained categories. For this fine-grained recognition task, we use ResNet-152 as the backbone network, which possesses a deeper network structure compared to ResNet-101. The optimizer is set

with a learning rate of 0.01, momentum of 0.99, and weight decay of $5 \times 10^{-4}$. The experimental results are shown in Figure 4, from which we observe that: 1) Our D-O2NC method converges faster than standard SGDM in terms of both training loss and accuracy; and 2) Our D-O2NC method achieves higher test accuracy than SGDM, further verifying the effectiveness and superior generalization ability of our algorithm.

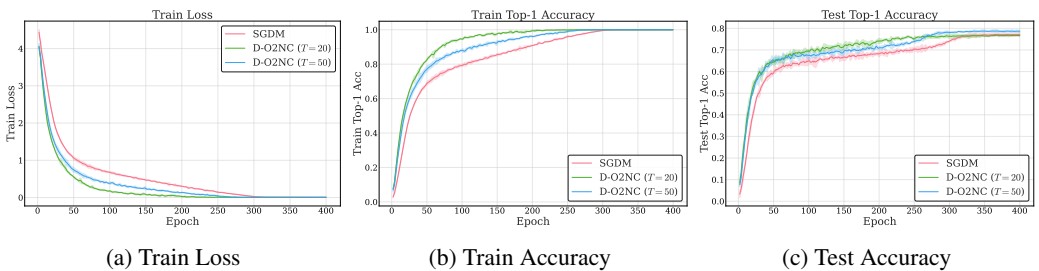

| (a) Train Loss | (b) Train Accuracy | (c) Test Accuracy |

Figure 4: Experimental results on CIFAR-100 with ResNet-152 network.

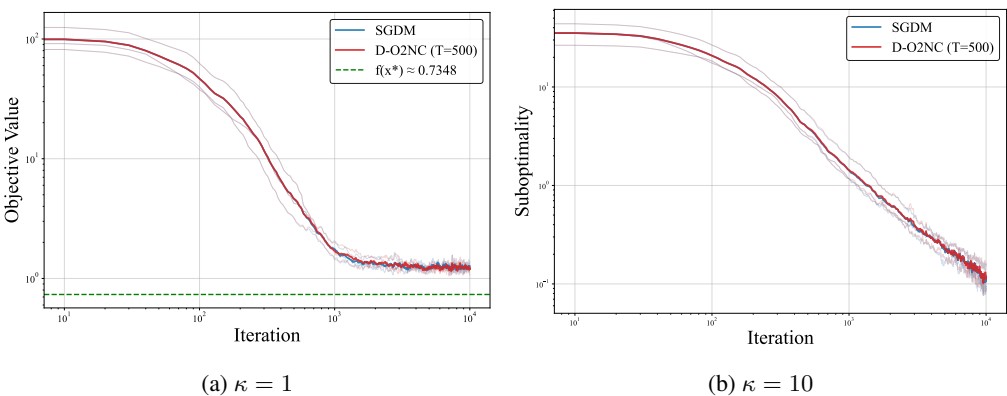

| (a) $\kappa = 1$ | (b) $\kappa = 10$ |

Figure 5: Experimental results on robust phase retrieval for two different condition numbers $\kappa$.

### E.2 EXPERIMENTS ON ROBUST PHASE RETRIEVAL

To further validate the effectiveness of our D-O2NC method for weakly convex problems, we conduct an additional set of experiments on the robust phase retrieval task (Duchi & Ruan, 2018). Given a set of $m$ measurement-amplitude pairs $\{x_i, y_i\}_{i=1}^m$, robust phase retrieval is typically formulated as the following composite minimization problem:

$$\mathcal{L}(w) = \frac{1}{m} \sum_{i=1}^m \left| (w^\top x_i)^2 - y_i \right|,$$

where $w$ denotes the unknown signal to be recovered, $x_i$ are the measurement vectors, and $y_i$ are the observed squared amplitude values. This loss function measures the discrepancy between the predicted squared inner-product magnitudes and the observed measurements. Clearly, it is of a composition form $h \circ c$, where $h(u) = |u|$ is Lipschitz continuous and $c(w) = (w^\top x_i)^2 - y_i$ is smooth. Thus, the loss function is weakly convex.

Following the related experimental setup of Mai & Johansson (2020), we generate a measurement matrix of size $300 \times 100$ as $X = QP$, where $P$ is diagonal matrix with condition number $\kappa \in \{1, 10\}$, and $Q$ consists of *i.i.d.* entries. The observation noise follows $\mathcal{N}(0, 25)$ and corrupts 20% of the data points. Both SGDM and D-O2NC use a momentum coefficient of 0.92 and learning rate of $2 \times 10^{-4}$, with a total of 10,000 iterations. Figure 5 plots the sub-optimality gap against the number of iterations for both values of $\kappa$. From these results, we observe that for both condition numbers, our method achieves performance comparable to SGDM under the same learning rate and momentum.

