# OpenReview forum: "Derandomized Online-to-Non-convex Conversion for Stochastic Weakly Convex Optimization"
_ICLR.cc/2026/Conference — ICLR 2026 Poster_

### Official Review · Reviewer_fXmg · 2025-10-31

**Soundness:** 3
**Presentation:** 2
**Contribution:** 2
**Rating:** 4
**Confidence:** 2

**Summary:**

The paper extends the Online-to-Non-Convex Conversion (O2NC) framework by introducing an online learning method that efficiently finds approximate stationary points for non-smooth, non-convex functions with optimal oracle complexity. It shows that when random interpolation or scaling is applied, O2NC becomes equivalent to the stochastic gradient descent with momentum algorithm commonly used in deep learning, providing a theoretical explanation for its effectiveness. Although it is generally impossible to remove randomness without losing optimality, the paper proves that for weakly convex functions, a deterministic version of O2NC can still achieve the same optimal rate when the weak convexity parameter is not too large. The authors also propose a momentum-restarted variant that improves progress when the algorithm is far from stationarity and demonstrate through experiments on ResNet and Vision Transformer networks that this version is both efficient and effective in practice.

**Strengths:**

The paper establishes that O2NC achieves an oracle complexity of O(deltaˆ(-1) epsilonˆ(-3)) for finding Goldstein (delta-stationary points) of non-smooth, non-convex functions. This result improves the state-of-the art complexity of this problem (e.g., compared to Cutkosky, 2019).

Another clear contribution is that theoretical analysis clearly characterizes the boundary between when randomness is necessary and when it is not. Using the impossibility results of Jordan et al. (2023), the paper explains why fully deterministic algorithms cannot achieve dimension-free rates in general non-convex settings. However, it provides a positive theoretical result showing that when the weak convexity parameter is smaller than O(deltaˆ(-1) epsilonˆ(-3)), randomness can be eliminated without losing optimality.

The experiments show that the proposed D-O2NC method improves upon the baseline SGDM algorithm. When applied to training deep neural networks on CIFAR-10 using ResNet-101 and ViT models, D-O2NC achieves faster convergence and slightly higher test accuracy, demonstrating the practical benefit of its periodic momentum restart mechanism.

**Weaknesses:**

The results are only for weakly convex functions, which is theoretically fine. But, are there many functions that are non-convex but weakly convex? In particular, if I understand correctly, the problem considered in experiments does not have this property, or at least the paper does not weakly convex, so I dont know, but my feeling is that all the interesting problems are convex. Thus it would be useful if there was better comparison with the state of the art for convex setup as well.

To my understanding, although the analysis is novel, the algorithm itself closely resembles existing momentum-based methods such as SGDM or restarted gradient methods. The contribution lies mainly in the theoretical interpretation rather than in a new optimization technique.

The experimental results are relatively weak and provide only limited empirical support for the proposed method. The evaluation relies on single runs without statistical analysis, making it difficult to assess the consistency or significance of the reported improvements. Moreover, there is little exploration of hyperparameters, such as restart frequency or learning rate, and no ablation study to clarify how these choices affect performance. The experiments also lack comparisons with other widely used optimizers beyond SGDM, which limits the context for evaluating the method’s advantages.

Moreover, the experimental section feels somewhat disconnected from the theoretical contributions of the paper, as it does not explicitly demonstrate or test the theoretical claims, such as optimal complexity or the benefits of weak convexity, in practice.

**Questions:**

I feel like there is some lack between the theory and practice. How would you for example use the theory to run the algorithm, e.g., hyper parameters.

Is it possible to determine or approximate in practice the step-size and other hyperparameter choices required by the theoretical analysis, and if so, how sensitive is the algorithm’s performance to deviations from these theoretically prescribed values?”

---

> ### Author Response · Authors · 2025-11-21
> **Response to Reviewer fXmg (I)**
>
> Thank you for your insightful review. We hope our following responses adequately address the identified weaknesses and questions, and we remain ready to address any further inquiries.
>
> > **Your comment:**  The results are only for weakly convex functions, which is theoretically fine. But, are there many functions that are non-convex but weakly convex?
>
> **Our response:**  Weakly convex functions are pervasive in practical applications and typically easy to identify.
> As noted in Line 152 (of the original submission), a common source of such functions is the composite form $f(x)=h(c(x))$, where $h$ is convex and $G$-Lipschitz continuous, and $c$ is a smooth mapping with a $L$-Lipschitz Jacobian. These composite functions are neither smooth nor convex but rather $GL$-weakly convex—they nicely interpolate between the smooth and convex settings.  A concrete example, as considered in ourexperimental study, is neural networks equipped with smooth activation functions (e.g., softplus and GeLU): their loss function take the composite form $f=h\circ c$ where $h$ is a convex predictor (e.g., cross-entropy loss) and $c$ is a smooth hierarchical feature map. Notably, softplus and GeLU typically match or outperform their non-smooth counterpart ReLU ([Clevert et al., 2016](https://arxiv.org/abs/1511.07289); [Xu et al., 2015](https://arxiv.org/abs/1505.00853)). For additional examples of weakly convex functions, we kindly refer the reviewer to [Davis & Drusvyatskiy (2019, Section 2.1)](https://arxiv.org/pdf/1803.06523) and [Asi & Duchi (2019, Section 2)](https://arxiv.org/pdf/1903.08619).
>
> > **Your comment:** In particular, if I understand correctly, the problem considered in experiments does not have this property, or at least the paper does not weakly convex, so I dont know, but my feeling is that all the interesting problems are convex. Thus it would be useful if there was better comparison with the state of the art for convex setup as well.
>
> **Our response:** There appears to be a minor misunderstanding here. To clarify upfront, a core motivation of our work is the derandomization of O2NC for SGDM recovery. Given ICLR’s strong focus on deep learning, omitting deep neural network experiments would be remiss. For this reason, our empirical study evaluates the performance of D-O2NC on neural network classification tasks using the CIFAR-10/100 datasets.
>
> Specifically, we adopt ResNet-101 and ViT architectures, pairing them with cross-entropy loss (a convex top-layer predictor) and replacing ReLU activations with GeLU (a smooth activation function). This design ensures the resulting objective functions are indeed *weakly convex*, aligning with the composite weak convexity structure we outlined earlier. While the convex setup you mentioned is certainly interesting, it is less relevant to the core focus of our work. We therefore plan to explore related experiments on convex functions as part of future research.
>
> We hope this clarifies both our experimental setup and the weak convexity of the tested objectives. Please feel free to let us know if you would like additional details on this aspect!
>
> > **Your comment:** To my understanding, although the analysis is novel, the algorithm itself closely resembles existing momentum-based methods such as SGDM or restarted gradient methods. The contribution lies mainly in the theoretical interpretation rather than in a new optimization technique.
>
> **Our response:** Your observation is entirely fair: while our D-O2NC algorithm bears resemblances to SGDM (up to momentum clipping or resetting), its core contribution indeed resides in the novel theoretical analysis and dedicated
> derandomization framework. Specifically, our work provides a rigorous deterministic foundation for recovering SGDM from the O2NC paradigm. This theoretical insight not only deepens the understanding of how SGDM behaves in weakly convex settings but also offers a principled way to derive and analyze such methods, rather than proposing an entirely new optimization technique.
>
> After our initial submission, we learned that the momentum-resetting technique has recently achieved experimental effectiveness in training large language models (LLMs) and deep reinforcement learning (deep RL) tasks ([Huang et al., 2025](https://openreview.net/pdf?id=L9eBxTCpQG), [Asadi et al., 2023](https://arxiv.org/pdf/2306.17833)). Notably, our convergence results on D-O2NC with restarted OGD thus provide a solid theoretical foundation for explaining the empirical success of the momentum-resetting technique.
>
> We agree that highlighting this distinction is important, and we have updated the manuscript to more explicitly emphasize the theoretical nature of our work, alongside its empirical validation.

---

> ### Author Response · Authors · 2025-11-21
> **Response to Reviewer fXmg (II)**
>
> >**Your comment:** The evaluation relies on single runs without statistical analysis, making it difficult to assess the consistency or significance of the reported improvements.
>
> **Our response:** As mentioned in the original submission (Lines 462-464), our empirical evaluation actually comprises three independent runs per experimental setting to ensure the reliability of results. To enhance transparency, we have updated the supplementary experimental results in Appendix E to include three independent runs consistently.
>
> > **Your comment:** Moreover, there is little exploration of hyperparameters, such as restart frequency or learning rate, and no ablation study to clarify how these choices affect performance.
>
> **Our response:** We would like to clarify that for each experimental setup, we have tested the performance of our D-O2NC method with varying restart frequency $T$, adopting identical learning rates and momentum coefficients to those used for SGDM (see Lines 457-463 of the original submission). To further alleviate this concern, we have supplemented additional experiments with diverse learning rate and momentum coefficient configurations. As detailed in the revised paper, the results demonstrate that D-O2NC consistently outperforms SGDM when both share the same hyperparameters.
>
> > **Your comment:** Moreover, the experimental section feels somewhat disconnected from the theoretical contributions of the paper, as it does not explicitly demonstrate or test the theoretical claims, such as optimal complexity or the benefits of weak convexity, in practice.
>
> **Our response:** Our experiments are not intended for theoretical verification; rather, their core goal is to assess the effectiveness of D-O2NC (equipped with restarted OGD, i.e., momentum-resetting SGDM) in training deep neural networks. The rate optimality and advantages of weak convexity underpinning our method have been rigorously established theoretically, rendering experimental validation unnecessary here. Instead, we focus on a more practical question: whether the theoretically derived momentum-resetting mechanism brings tangible performance gains to SGDM in deep learning, bridging theory with real-world utility.
>
> > **Your comment:** I feel like there is some lack between the theory and practice. How would you for example use the theory to run the algorithm, e.g., hyper parameters.
>
> **Our response:** We emphasize that our hyperparameter choices are not arbitrary, but rather closely aligned with the theoretical settings outlined in Corollary 1 and Corollary 2. For instance, with precisions $\delta=\epsilon=\mathcal{O}(N^{-1/4})$, Corollary 1 prescribes  $T=\mathcal{O}(N^{-1/2})$ (restart frequency), $\gamma^{-1}=\mathcal{O}(N^{-1/2})$ (learning rate), and $1-\beta= 1-\mathcal{O}(N^{-1/2})$ (momentum coefficient).  Following your comment, we have added a remark below Corollary 1 that explicitly states the hyperparameter settings derived from our theory.
>
> > **Your comment:**  Is it possible to determine or approximate in practice the step-size and other hyperparameter choices required by the theoretical analysis, and if so, how sensitive is the algorithm’s performance to deviations from these theoretically prescribed values?”
>
> **Our response:** Yes, the hyperparameters proposed by our theory are indeed computationally tractable in practice. For instance, as detailed in our prior response, the theoretical hyperparameter choices outlined in Corollary 1 are uniquely determined by the iterate budget $N$ and desired precisions $\delta,\epsilon$, which can typically be specified by users in practical applications. The same logic applies to Corollary 2, where hyperparameters are similarly derived from user-defined inputs aligned with theoretical requirements.
>
> Concerning the sensitivity of algorithm performance to theoretical hyperparameters, we have supplemented additional experiments with varying learning rates and momentum coefficients. As detailed in the revised paper, the results demonstrate that D-O2NC performs stable under a wide range of hyperparameters around the theoretical values.
>
> Should our response address your concerns, we would appreciate you revisiting the contributions of our work and reconsidering your evaluation.

---

### Official Review · Reviewer_xFGb · 2025-10-31

**Soundness:** 3
**Presentation:** 3
**Contribution:** 3
**Rating:** 8
**Confidence:** 4

**Summary:**

This paper studies the Online-to-nonconvex conversion (O2NC) under weakly convex scenario. While the original O2NC has general update form $w_n = w_{n-1} + s_n\Delta_n$, where $\Delta_n$ denotes the update vector and $s_n$ is a random scaling or interpolation, this paper proposes derandomized O2NC (D-O2NC) with deterministic update form $w_n = w_{n-1} + \Delta_n$. The core difference is that O2NC relies on the identity $\mathbb{E}\_{s_n } [R(w_n) - R(w\_{n-1})] = \mathbb{E}[ \langle \nabla R(w_n), \Delta_n\rangle]$ from random scaling, while D-O2NC relies on the weak convexity identity $R(w_n) - R(w_{n-1}) \le \langle \nabla R(w\_n), \Delta\_n\rangle + \frac{\rho}{2}\\|\Delta_n\\|^2$, where $\rho$ denotes the weak convexity constant. Consequently, upon substituting online learners with proper regret bound on the quadratic loss $f_n(\Delta) = \langle g\_n, \Delta\rangle + \frac{\rho}{2}\\|\Delta\\|^2$, the resulting D-O2NC optimizer achieves good convergence guarantee. In particular, this paper considers two such online learners, clipped OGD and unclipped OGD with periodic restarting, both achieving sub-linear regret on the aforementioned quadratic losses and recovering variants of the popular SGDM optimizer upon substituting into D-O2NC. Finally, this paper also provides empirical experiments on cifar10 with resnets and ViT models and shows that D-O2NC outperforms standard SGDM.

**Strengths:**

- The weakly convex scenario is rarely studied in the line of O2NC research, and this paper provides new results along this line. These results are novel, and they help to better understand the performance of O2NC under different settings.
- In the convergence rate, I find it interesting that the weak convexity constant $\rho$ is only in the non-dominant terms. In other words, the convergence rate of derandomized O2NC matches that of O2NC as long as weak convexity is bounded by $O(\delta^{-1}\epsilon^{-1})$, even though there is no random scaling.
- Compared to original O2NC, derandomized O2NC better aligns with the practical implementation of popular optimizers such as SGDM, in which the random scaling is typically absent.
- Empirical results on cifar-10 with resnets and ViT demonstrates better practical performance of D-O2NC compared to vanilla SGDM.
- Overall, I find this paper well-written. The presentation of the main results and corresponding discussions are clear and easy to follow.

**Weaknesses:**

The discussion of the lower bound on the convergence rate of Goldstein stationary point of a weakly convex loss is missing. Although $O(\delta^{-1}\epsilon^{-3})$ is the minimax optimal rate for the general non-convex losses, it's unclear whether it remains tight for weakly convex functions. In fact, in the discussion section about the connection to Moreau envelope (e.g. Appendix D), it seems to me that the previously achieved rates on Moreau envelope translates to better rates on Goldstein stationary point. Could the authors confirm if I understand correctly, and provide some insights on the lower bound?

**Questions:**

- Regarding the definition of regulated Goldstein stationary, my understanding is that this definition makes it more convenient for the convergence analysis, but does not include more optimizer algorithms (unlike the relaxed Goldstein stationary definition which relaxes the deterministic radius $\delta$ to a stochastic variance bound). This is further convinced by the equivalence relation between standard Goldstein stationary and regulated version (Lemma 1). Am I understanding correctly?

---

> ### Author Response · Authors · 2025-11-21
> **Response to Reviewer xFGb**
>
> Thank you for your insightful review and appreciation of our work.
>
> > **Your comment:**  Although $\mathcal{O}(\delta^{-1}\epsilon^{-3})$  is the minimax optimal rate for the general non-convex losses, it's unclear whether it remains tight for weakly convex functions.
>
> **Our response:**  We confirm that the $\mathcal{O}(\delta^{-1}\epsilon^{-3})$ rate is indeed tight for weakly convex functions. As highlighted in Remark 2 (Lines 319-323 of the original submission), the key insight is that the $\mathcal{O}(\delta^{-1}\epsilon^{-3})$ rate is known to be unimprovable [(Cutkosky et al., 2023, Theorem 18)](https://openreview.net/pdf?id=GimajxXNc0) for all $\epsilon \le \mathcal{O}(\delta)$—a result that holds even for smooth functions, let alone their superclass of weakly convex functions. To be more precise, [Cutkosky et al. (2023, Proposition 14)](https://openreview.net/pdf?id=GimajxXNc0) showed that a $(\delta,\epsilon)$-stationary point of an $H$-smooth function yields a $(\epsilon+H\delta)$-stationary point, which generally requires  $\Omega (H(\epsilon + H\delta)^{-4})$ oracle queries to identify by stochastic first-order algorithms. Then, by setting $H=\epsilon/\delta$, we find that  $\Omega (\delta\epsilon^{-3})$ oracle queries are necessary to produce a $(\delta,\epsilon)$-stationary point. Per your comment, we have supplemented a more detaild discussion on the tightness of our result for weakly convex functions in the revised paper.
>
> > **Your comment:**  In fact, in the discussion section about the connection to Moreau envelope (e.g. Appendix D), it seems to me that the previously achieved rates on Moreau envelope translates to better rates on Goldstein stationary point. Could the authors confirm if I understand correctly, and provide some insights on the lower bound?
>
> **Our response:** We greatly appreciate your careful review of the appendix and valuable comments. There appears to be a minor misunderstanding here: the prior rates for the Moreau envelope actually *do not* translate to better rates on the Goldstein stationary point. Indeed, as discussed in Remark 9 in Appendix D, the $\epsilon$-statinoary point of the Moreau envelope of a $\rho$-weakly convex function $f$ implies a $(\epsilon/(2\rho), \epsilon)$-stationary point of $f$. Therefore, by specifically choosing $\delta= \epsilon/(2\rho)$, the best-known $\mathcal{O}(\rho\epsilon^{-4})$ rates for the Moreau envelope translate to $\mathcal{O}(\delta^{-1}\epsilon^{-3})$ rates on Goldstein stationarity which are actually *consistent with our results*. However, a limitation of translating rates via the setting $\delta= \epsilon/(2\rho)$ is that it excludes the range of relatively large $\delta$ (e.g., $\delta=\sqrt{\epsilon}$, a choice critical for sharper rates in second-order smooth functions), as $\rho$ is typically lower bounded by a constant. We have updated Remark 9 (as Remark 10 in the revised paper) to avoid confusion.
>
> Regarding your question about the lower bounds, please kindly refer to our prior response for a detailed reply.
>
> > **Your comment:** * Regarding the definition of regulated Goldstein stationary, my understanding is that this definition makes it more convenient for the convergence analysis, but does not include more optimizer algorithms (unlike the relaxed Goldstein stationary definition which relaxes the deterministic radius $\delta$ to a stochastic variance bound). This is further convinced by the equivalence relation between standard Goldstein stationary and regulated version (Lemma 1). Am I understanding correctly?
>
> **Our response:** Yes, your interpretation is correct! Our regulated Goldstein stationarity is indeed defined to streamline convergence analysis, while retaining the original almost sure $\delta$-ball constraint (rather than relaxing it to a stochastic $\delta^2$-variance constraint). We greatly appreciate your precise grasp of this key distinction, which aligns perfectly with our original intent.

---

### Official Review · Reviewer_FDFs · 2025-10-31

**Soundness:** 4
**Presentation:** 4
**Contribution:** 3
**Rating:** 8
**Confidence:** 4

**Summary:**

This paper provides an algorithm for finding Goldstein stationary points of weakly convex functions using the "online-to-non-convex" framework.
The resulting algorithm does not require randomization, which is required for previous analyses that do not require weak convexity.

A few experimental results are provided using cifar10.

**Strengths:**

Removing the randomization is intuitively a desirable behavior, and the present result provides a more general way to obtaining convergence rates without randomization. In particular, it strictly generalizes prior results finding stationary points of smooth or second-order smooth objectives.

**Weaknesses:**

The most obvious weakness is that the results for finding stationary points of the Moreau envelope are suboptimal.

**Questions:**

For the experiments, what is meant by learning rate and momentum parameters? How are they translated to the new algorithms? Is the learning rate the learning rate of the inner online learner or something else?

The resulting algorithms look rather similar to SGD with momentum. Indeed, without the clipping value, option 1 appears to literally be SGD with momentum. It looks like the algorithm of Cutkosky & Zhang does not require clipping through use of an extra regularization - would that be applicable here? In such a case, the analysis of Mai & Johanansson would also apply to this algorithm and so both kinds of convergence would be achievable.

---

> ### Author Response · Authors · 2025-11-21
> **Response to Reviewer FDFs**
>
> Thank you for your insightful review and positive evaluation of our work.
>
> > **Your comment:** The most obvious weakness is that the results for finding stationary points of the Moreau envelope are suboptimal.
>
> **Our response:**  We agree that our results for bridging Goldstein stationarity to Clarke stationarity of the Moreau envelope are suboptimal. Your insight aligns closely with our own thoughts: these findings, offered only as a side contribution in the appendix, were not designed to be a primary outcome. We greatly value your insights, which will guide our future work to refine the analysis and narrow the suboptimality gap.
>
> > **Your comment:** For the experiments, what is meant by learning rate and momentum parameters? How are they translated to the new algorithms? Is the learning rate the learning rate of the inner online learner or something else?
>
> **Our response:** The learning rate and momentum parameters reported in our experiments  correspond specifically to those of the recovered SGDM optimizer. More precisely, as stated in Lines 291-297 (and Lines 344-347) of the original submission, the recovered SGDM learning rate $\gamma^{-1}$ corresponds to the inverse of regularization strength of the quadratic losses $\langle \hat g_n, \cdot\rangle + \frac{ \gamma}{2}\|\cdot\|^2$ in our D-O2NC algorithm, and its momentum coefficient is given by $1-\beta=1-\eta\gamma$ where $\eta$ is the  learning rate of the OGD module in D-O2NC.
>
> > **Your comment:**  It looks like the algorithm of Cutkosky & Zhang does not require clipping through use of an extra regularization - would that be applicable here? In such a case, the analysis of Mai & Johanansson would also apply to this algorithm and so both kinds of convergence would be achievable.
>
> **Our response:** We are confident that the approach of employing the exponentiated loss combined with model exponential moving average (EMA), as proposed by [Zhang & Cutkosky (2024)](https://arxiv.org/pdf/2405.09742), is fully compatible with our D-O2NC framework and can be effectively integrated to achieve better recovery of SGDM. Notably, the extra regularization proposed by [Zhang & Cutkosky (2024)](https://arxiv.org/pdf/2405.09742) is unnecessary for our framework, as our loss function already includes a quadratic regularization component. We further agree that both Goldstein stationarity and Clarke stationarity (of the Moreau envelope) are achievable for SGDM under weak convexity. We intend to explore this promising potential as a future research direction for D-O2NC.

---

### Official Review · Reviewer_11DY · 2025-11-11

**Soundness:** 4
**Presentation:** 3
**Contribution:** 3
**Rating:** 4
**Confidence:** 3

**Summary:**

This paper proposes a derandomized version of the Online-to-Non-Convex Conversion (O2NC) framework for stochastic weakly convex optimization. Two options are given, one is clipped SGDM, and the other is restarted SGDM. The derandomization part is novel but the the weakly convexity condition seems to be strong for this problem class.

**Strengths:**

1. This paper proposes a deterministic framework that achieves the optimal complexity, showing that this is doable for weakly convex functions even not doable for general non-convex non-smooth functions.
2. Weaker convexity is allowable when the desired error is small.

**Weaknesses:**

1.  It's not clear under the weakly convex condition, what's the technical challenges addressed in this work.
2. Lack of examples of weakly convex functions and related numerical experiments. It would be very helpful if there are some simple function constructions to understand these bounds.

**Questions:**

1. Is weak convexity necessary for achieving optimal rates with deterministic algorithms?
2. Can you elaborate the hyperparameter tuning of the experiment section? Is it possible that SGDM converges slower due to suboptimal hyper-parameters?

---

> ### Author Response · Authors · 2025-11-21
> **Response to Reviewer 11DY (I)**
>
> Thank you for your insightful review. We hope our following responses adequately address the identified weaknesses and questions, and we remain ready to address any further inquiries.
>
> > **Your comment:** It's not clear under the weakly convex condition, what's the technical challenges addressed in this work.
>
> **Our response:**  First and foremost, we would like to reemphasize that identifying the derandomization potential of the original O2NC framework under weak convexity constitutes the most fundamental contribution of our work. With this in mind, there are two critical technical hurdles that must be addressed to apply O2NC under weak convexity, as detailed below.
>
> 1. Weakly convex functions are generally *not* differentiable everywhere, and thus fail to satisfy the key well-behavedness condition required by O2NC. To tackle this challenge, we propose to utilize the identity $R(w_n) - R(w_{n-1}) \le \mathbb{E}\left[\langle \hat g_n, \Delta_n\rangle + \frac{ \gamma}{2}\|\Delta_n\|^2\right]$ for a $\rho$-weakly convex function and $\gamma\ge \rho$, and choose the increments $\Delta_n$ in an online manner to make the quadratic regret $\sum_{n=1}^N \langle \hat g_n, \Delta_n\rangle + \frac{ \gamma}{2}\|\Delta_n\|^2$ as low as possible, such that the objective value gap $R(w_N)-R(w_0)$ can be well upper bounded.
> 2. The clipped online gradient descent (OGD) employed in the original O2NC framework constrains the increments $\Delta_n$ to stay inside a sufficiently small ball. This constraint, however, may be overly conservative and hinder effiient model updates. To address this challenge, we further develop a novel periodically reset OGD procedure for O2NC, which allows for more progressive update especially when the iterates are far from stationary.
>
> > **Your comment:** Lack of examples of weakly convex functions and related numerical experiments. It would be very helpful if there are some simple function constructions to understand these bounds.
>
> **Our response:**
>
> 1. **Examples of weakly convex functions.** The class of weakly convex functions is rich and generally easy to recognize in applications. As mentioned in the original submission (Line 152), one common source is the composite form $f(x)=h(c(x))$ where $h$ is a convex and $L$-Lipschitz continuous function, and $c$ is a smooth mapping with a $\beta$-Lipschitz Jacobian. These composite functions are neither smooth nor convex, but rather $L\beta$-weakly convex.  As a concrete example—consistent with the experimental study in our paper—neural networks equipped with smooth activation functions like softplus and GeLU are indeed weakly convex: the loss function is of composite form $f=h\circ c$, where $h$ is a convex top-layer predictor (e.g., cross-entropy loss) and $c$ is a smooth hierarchical feature mapping. Notably, softplus and GeLU typically match or exceed the performance of their non-smooth counterpart ReLU ([Clevert et al., 2016](https://arxiv.org/abs/1511.07289); [Xu et al., 2015](https://arxiv.org/abs/1505.00853)). For more examples of weakly convex functions (e.g., robust phase retrieval, biconvex compressive sensing), we kindly refer the reviewer to the work of [Davis & Drusvyatskiy (2019, Section 2.1)](https://arxiv.org/pdf/1803.06523) and [Asi & Duchi (2019, Section 2)](https://arxiv.org/pdf/1903.08619).
>
> 2. **Related numerical experiments.**  Since one of the key motivations of our work is to derandomize O2NC for recovering SGDM, and as ICLR centers on deep learning, omitting deep neural network experiments would be an oversight. Therefore, in our empirical study, we test the performance of our D-O2NC method for training neural networks for classification tasks over the CIFAR-10/100 datasets. We use the ResNet-101 and ViT architectures in which we replace the ReLU activations with GeLU (see Lines 455-456 of the original submission) and use cross-entropy loss so that the overall objective function is indeed weakly convex.
>
>    As per your comment, we have further elaborated on the discussions regarding examples of weakly convex functions and added a new set of numerical experiments on *robust phase retrieval* in the revised manuscript.
>
>
> > **Your comment:** Is weak convexity necessary for achieving optimal rates with deterministic algorithms?
>
> **Our response:**  The necessity of weak convexity for achieving optimal rates without randomization is a very thrilling open question. Given that the weak-convexity parameter $\rho$ can scale as large as $\mathcal{O}(\delta^{-1}\epsilon^{-1})$ without compromising rate optimality, we conjecture there remains substantial potential to extend the boundary of function class beyond the weakly convex setting. We leave a comprehensive understanding of this open problem to future work.

---

> ### Author Response · Authors · 2025-11-21
> **Response to Reviewer 11DY (II)**
>
> > **Your comment:** Can you elaborate the hyperparameter tuning of the experiment section? Is it possible that SGDM converges slower due to suboptimal hyper-parameters?
>
> **Our response:** We would like to highlight that our D-O2NC with periodically restarted OGD corresponds to a momentum-resetting variant of SGDM. Compared to SGDM, our restarted D-O2NC has only one additional key hyperparameter $T$ (restart period) other than the learning rate and momentum coefficient. Throughout the numerical study, our D-O2NC shares the exact same learning rate and momentum coefficient as SGDM, only with $T$ varied during testing.Thus, the slower convergence of SGDM is by no means due to suboptimal hyperparameter choices!
>
> To ease this concern, we have further conducted additional experiments under varying configurations of the learning rate and momentum coefficient. The results, which are available in the revised paper, show that D-O2NC consistently outperforms SGDM under the same configuration of hyperparameters.
>
>
> Should our response adequately address your concerns, we would greatly appreciate it if you could revisit our paper’s contributions and reconsider your evaluation.

---

### Author Response · Authors · 2025-11-21
**General response**

We are grateful to all reviewers for their insightful reviews. Guided by their constructive comments, we have carefully revised the manuscript, with a summary of key changes provided as follows:

1. We have further elaborated on the examples of weakly convex functions (see Section 2.1) and added a new set of numerical experiments on the robust phase retrieval problem, which is typically weakly convex (see Appendix E.2).
2. In Appendix E.1, we present additional experiments on neural networks, covering diverse configurations of the learning rate and momentum coefficient for all considered algorithms. Furthermore, we have updated the supplementary experimental results to consistently include three independent runs to ensure the reliability of our findings.
3. In Remark 2, we have refined the discussion on the tightness of our result for weakly convex functions.
4. In Remark 4, we provide an additional discussion on how to prescribe the key hyperparameters of our method based on Corollary 1, using user-specified quantities such as the iterate budget and stationarity precisions.
5. In Remark 11, we further clarify a limitation of translating the convergence rates from the Clarke stationarity of the Moreau envelope to the Goldstein stationarity of the original objective function.
6. We have corrected the typos and other minor issues noted in the reviews.

All major revisions are highlighted in red text. We sincerely hope that the concerns raised by the reviewers have been satisfactorily addressed in both the revised manuscript and our point-by-point responses.

---

### Meta-Review · Area_Chair_1xPs · 2025-12-27

**Summary:**

This paper proposes **Derandomized Online-to-Nonconvex Conversion (D-O2NC)**, a deterministic variant of the O2NC framework for stochastic optimization of weakly convex objectives. Unlike classical O2NC, which relies on random interpolation or scaling and deviates from practical momentum methods, D-O2NC removes randomness by exploiting weak convexity, using deterministic updates $w_n = w_{n-1} + \Delta_n.$

The analysis leverages the weak convexity inequality $R(w_n) - R(w_{n-1}) \le \langle \nabla R(w_n), \Delta_n \rangle + \frac{\rho}{2}\| \|\Delta_n\| \|^2,$  and frames the update of increments as online learning over quadratic losses. This yields the optimal oracle complexity $O(\delta^{-1}\varepsilon^{-3})$ for finding Goldstein $(\delta,\varepsilon)$-stationary points, allowing the weak-convexity parameter to scale as $\rho = O(\delta^{-1}\varepsilon^{-1})$ without degrading the dominant rate.

Two concrete instantiations are studied: a clipped OGD variant and a periodically restarted OGD variant, the latter closely corresponding to a momentum-restarted version of SGDM. Empirical results on CIFAR-10 with ResNet and Vision Transformer models show that D-O2NC consistently outperforms standard SGDM in both convergence and generalization.

**Reviewer Concerns:**

Below, I briefly summarize the reviewers' concerns and explain how the authors addressed them (in brackets).

**Reviewer 11DY.**

- Not clear what the technical challenges are (explained by authors in the rebuttal)
- Request to provide examples of weakly convex functions (not a concern, but more a question; fully addressed by the authors)

**Reviewer FDFs.**

- The suboptimality of the results for finding stationary points of the Moreau envelope (not the main focus of the paper, as the authors pointed out)
- Several technical questions about experimental details and applicability of the approach from Cutkosky & Zhang paper (addressed by the authors)

**Reviewer xFGb.**

- Tightness of the complexity bound for weakly convex functions (indeed tight, as explained by the authors in the rebuttal)
- Rates on the Moreau envelope translate to better rates for Goldstein stationarity (misunderstanding; addressed by the authors)

**Reviewer fXmg.**

- Question about the relevance of the weakly-convex functions (fully addressed by the authors)
- Concern about the contribution nature (addressed by the authors in the rebuttal and in the revised manuscript)
- Concerns about the experiments and connection with the theory (new details were added; the paper is mostly theoretical)

**Reviewer Scores:**

**Reviewer 11DY.** The concerns were minor and required just a few clarifications on the technical challenges and examples of weakly convex functions. The authors fully addressed these concerns, so I assume that the score after the rebuttal would be 6 or 8.

**Reviewer FDFs.** The review was already positive, though the authors' clarifications are useful. I think the reviewer would keep the score of 8.

**Reviewer xFGb.** The review was already positive, though the authors' clarifications are useful. I think the reviewer would keep the score of 8.

**Reviewer fXmg.** From what I see, the concerns were relatively minor. The reviewer raised several questions, and the authors addressed them in the rebuttal. I think the reviewer would increase the score to 6.

**That is, I assume that after the rebuttal, the reviewers would acknowledge that their concerns are fully resolved. Given this and the overall quality of the paper, I recommend acceptance.**

---

### Decision · Program_Chairs · 2026-01-26

Accept (Poster)